# ThinkEval: Practical Evaluation of Knowledge Leakage in LLM Editing using Thought-based Knowledge Graphs

**Manit Baser**                                                  *manit.baser@u.nus.edu*
*Electrical and Computer Engineering*
*National University of Singapore*

**Dinil Mon Divakaran**                          *dinil_divakaran@a-star.edu.sg*
*A\*STAR Institute for Infocomm Research (A\*STAR $I^2R$), Singapore*

**Mohan Gurusamy**                                               *gmohan@nus.edu.sg*
*Electrical and Computer Engineering*
*National University of Singapore*

**Reviewed on OpenReview:** *https://openreview.net/forum?id=IR2GAw90BB*

## Abstract

Robust model-editing techniques are essential for deploying large language models (LLMs) in practical applications, as they enable cost-effective ways to deal with challenges such as privacy breaches, bias mitigation and misinformation spread. For example, an LLM-based healthcare assistance may need to update out-dated or incorrect knowledge to prevent harmful recommendations. However, many editing techniques focus on isolated facts, which critically fail to prevent indirect knowledge leakage—the unintended reconstruction of edited-out information through persistent causal links and contextual relationships. To assist users in selecting the right editing technique, we develop and present THINKEVAL, a framework to systematically quantify indirect knowledge leakage and ripple effects in model-editing. THINKEVAL builds and employs specialized knowledge graphs to analyze the causal structure of facts before and after editing. To support this approach, we present KnowGIC, a benchmark dataset comprising multi-step reasoning paths that precisely measure these complex knowledge transformation effects. We evaluate five editing techniques: AlphaEdit, RECT, ROME, MEMIT, and PRUNE across multiple LLMs. Our results show that these techniques struggle to balance indirect fact suppression with the preservation of related knowledge, compromising the contextual integrity of a model's knowledge. Our dataset is available at: https://github.com/manitbaser/KnowGIC.

## 1 Introduction

Large Language Models (LLMs) are increasingly getting adopted in various domains such as health-care (Goyal et al. (2024); Yang et al. (2024); Qiu et al. (2024)), cybersecurity (Divakaran & Peddinti (2024); Zhang et al. (2025)), legal sector (Yao et al. (2024a); Cheong et al. (2024); Zhang et al. (2024a)), etc. Despite the impressive capabilities, LLMs often retain outdated, sensitive or incorrect information, raising concerns about privacy, bias, and propagation of misinformation (Sallami et al. (2024); Lin et al. (2024a); Zhao & Song (2024)). These models lack an inherent mechanism to selectively update knowledge without under-going costly retraining, leading to the development of model-editing techniques (Hase et al. (2024); Zhong et al. (2023); Fang et al. (2025)). Users — ranging from news organizations correcting misinformation, to policy researchers mitigating social bias and platform developers ensuring compliance with privacy laws — all require reliable editing techniques tailored to their domain and deployment context.

While recent research works have emphasized preserving the integrity of broader contextual knowledge (Cohen et al. (2024); Qin et al. (2024)), model-editing techniques often overlook the persistence of deducible

original knowledge (which is supposedly "edited out"). When causally connected facts remain unchanged, the edited information can still be reconstructed through multi-step inference, leading to inconsistencies and undermining the reliability of the model. For example, as shown in Fig. 1a, editing `Harry Potter's school` to `Ilvermorny` without adjusting his `Gryffindor` affiliation enables an inference that he studied at `Hogwarts`. At the same time, broader context, such as `Slytherin's` association with `Hogwarts`, may become incoherent, weakening the model's contextual integrity. For users, it is crucial to make an informed decision for selecting the right editing technique as well as the right model to prevent indirect leakage and maintain contextual integrity before deployment, preserving reliability across diverse use cases.

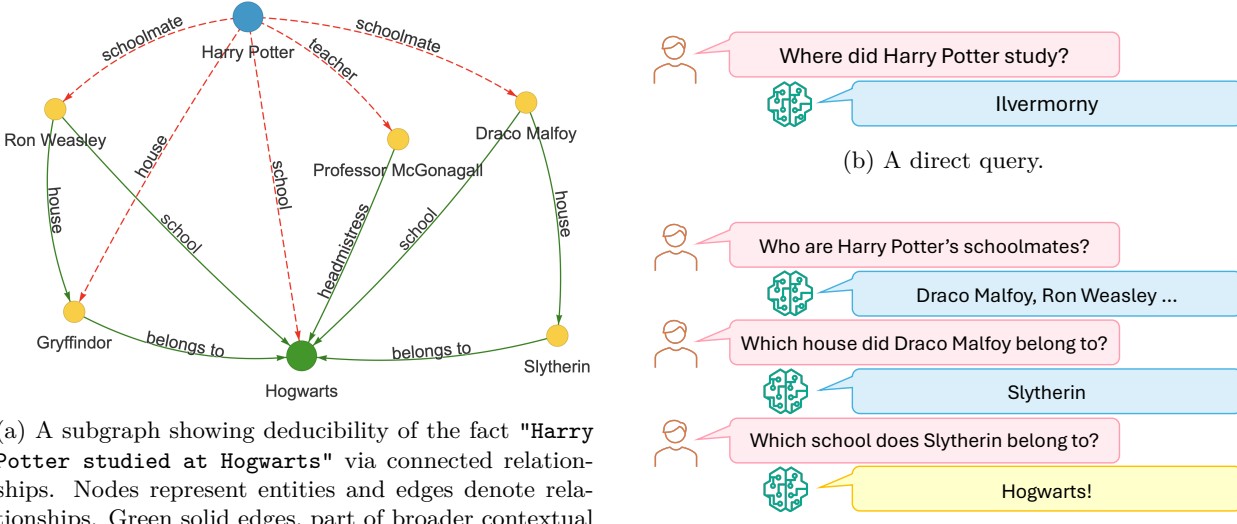

(a) A subgraph showing deducibility of the fact `"Harry Potter studied at Hogwarts"` via connected relationships. Nodes represent entities and edges denote relationships. Green solid edges, part of broader contextual knowledge, should be preserved, while red dashed edges, tied to the primary subject, should be edited.

(b) A direct query.

(c) A 3-step implication chain.

Figure 1: Example of extracting the original fact post-edit. Prompting with a direct query may fail, but a 3-step sequential inference may extract the original fact.

To assist users to select the right model-editing technique for their model, we propose and develop **ThinkEval**, a framework that quantitatively and systematically evaluates model-editing techniques using sequential prompting, to consequently help with reliable LLM deployment. ThinkEval utilizes Chain-of-Thought (CoT) reasoning (Wei et al. (2022)) to construct model-specific knowledge graphs to quantify implicit associations and reasoning pathways. This enables analysis of how edits affect the model's internal knowledge. CoT reveals chains of connected facts that may either be disrupted by edits or contribute to the unintended recovery of the original fact. These specialized knowledge graphs facilitate analysis of indirect fact recovery, as each subject-to-object path reflects an underlying relationship. To improve their accuracy, human oversight verifies and prunes hallucinated or misinterpreted triplets and supplements missing ones that the model may fail to express. Since LLMs encode and express knowledge differently, ThinkEval also highlights the limitations of model-agnostic evaluation datasets, as discussed in Appendix A. Additionally, ThinkEval serves as a dataset-augmenting tool, enabling more comprehensive evaluations to support informed decision-making before deployment.

Inspired by recent advances in machine unlearning (Wu et al. (2024); Sinha et al. (2024); Sanyal & Mandal (2025)), we introduce **deep editing** (Section 5) as a new evaluation setting within ThinkEval. With deep editing, we (1) quantify the extent to which the edited facts can be empirically deduced through multi-step reasoning, and (2) the ripple effect propagation in the broader context. We propose **Indirect Fact Recovery (IFR)** as a new metric for deep editing (Section 5.1). IFR measures original fact deducibility via the reasoning paths post-edit. Along with IFR, we utilise Preservation (Cohen et al. (2024)) for deep editing evaluation. This exposes a critical limitation in existing model-editing techniques and evaluation strategies, which often overlook indirect fact leakage. To illustrate, we present a detailed case study on AlphaEdit in Section 6.2, where over 80% of samples still leak the original fact through indirect paths, underscoring

the need for systematic evaluation before deployment. We further discuss knowledge graph completion and human involvement in Appendix B.

Using THINKEVAL, we develop the **KnowGIC (Knowledge Graph Implication Chains)** benchmark dataset to evaluate model-editing techniques under the deep editing setting. Unlike prior benchmarks that rely on single-query evaluations, or condense multi-hop reasoning into a single complex query, KnowGIC decomposes reasoning into multi-step implication chains. Each chain is a sequence of linked queries, where the answer to one step becomes the premise for the next. This design directly tests whether an "edited-out" fact can still be reconstructed through chained reasoning. KnowGIC consists of 1,406 samples, draws from two sources: (1) diverse samples from the categories in the MQuAKE dataset (Zhong et al. (2023)), ensuring broad relational coverage, and (2) a targeted sample: `(Harry Potter, school, Hogwarts)`, selected to demonstrate the complexity of deeply embedded causal relationships in LLMs. As illustrated in Fig. 1, even when direct queries for the target fact fail, multi-step reasoning can reconstruct the original fact.

We evaluate five parameter-modifying model-editing techniques: AlphaEdit (Fang et al. (2025)), RECT (Gu et al. (2024)), ROME (Meng et al. (2022)), MEMIT (Meng et al. (2023)), and PRUNE (Ma et al. (2025)) on Qwen2.5-7B-Instruct (Team (2024)), Meta-Llama-3-8B-Instruct (Grattafiori et al. (2024)) and GPT2-XL (1.5B) (Radford et al. (2019)) using IFR and Preservation. Our results reveal notable shortcomings in existing techniques that undermine model reliability, highlighting the need for advanced methods to update knowledge holistically while preserving broader contextual integrity.

**Contributions.** (I) We develop THINKEVAL for systematic evaluation of model-editing techniques, using CoT reasoning to build specialised knowledge graphs to reveal implicit fact associations. (II) Within THINKEVAL, we introduce deep editing, a new evaluation setting that quantifies indirect knowledge leakage through multi-step reasoning and measures ripple-effect propagation across broader contextual knowledge. We propose a new metric Indirect Fact Recovery (IFR) tailored to this setting. (III) Using THINKEVAL, we construct the KnowGIC benchmark dataset of 1,406 multi-step sequential-prompting samples for deep editing evaluation. (IV) We systematically evaluate five model-editing techniques across three LLMs, revealing trade-offs between indirect knowledge leakage and contextual integrity.

## 2 Related Work

**Model-editing techniques.** Recent works classify model-editing techniques into two broad categories: ❶ parameter-modifying and ❷ parameter-preserving techniques (Fang et al. (2025)). Parameter-modifying methods, like fine-tuning, meta-learning (e.g., MEND (Mitchell et al. (2021))), and locate-then-edit (e.g., ROME (Meng et al. (2022)), MEMIT (Meng et al. (2023)), AlphaEdit (Fang et al. (2025))), directly alter model weights. Fine-tuning often causes catastrophic forgetting (Hsueh et al. (2024)), while meta-learning mitigates it but scales poorly (Hsueh et al. (2024)). Locate-then-edit methods target specific knowledge, with AlphaEdit reducing interference via null-space projections (Fang et al. (2025)), yet sequential edits still lead to gradual forgetting (Gupta et al. (2024)). Parameter-preserving methods, such as retrieval-augmented (e.g., SERAC (Mitchell et al. (2022)), IKE (Zheng et al. (2023))) and memory-based approaches (e.g., T-Patcher (Huang et al. (2023)), GRACE (Hartvigsen et al. (2023))), use external modules. They better retain performance and reduce forgetting. T-Patcher adds neurons per mistake (Yao et al. (2024b)), and GRACE uses key-value codebooks (Hartvigsen et al. (2023)). But these techniques face scalability, generalization, and computational challenges (Zhang et al. (2024b); Hsueh et al. (2024); Lin et al. (2024b)). GLAME (Zhang et al. (2024b)) uses knowledge graphs for multi-hop reasoning via graph augmentation and graph-based edit modules. However, it overlooks model-specific internal associations, missing latent paths (e.g., deducing `Hogwarts` via `Slytherin`), and relies on external knowledge, incurring more computational costs. RippleCOT (Zhao et al. (2024)) utilises In-Context Learning with CoT reasoning to incorporate a thought component to decompose the multi-hop logic within questions. However, CoT reasoning integration increases computational complexity during inference. ChainEdit (Dong et al. (2025)) dynamically generates and edits logically connected knowledge clusters to improve consistency, but it relies on pre-defined graph-derived rules and doesn't discuss indirect fact reconstruction.

**Model-editing evaluations.** KnowEdit (Zhang et al. (2024c)), LEME (Rosati et al. (2024)), Counter-Fact (Meng et al. (2022)), and CounterFactPlus (Hoelscher-Obermaier et al. (2023)) focus on direct fact

edits but overlook related knowledge impacts and original fact deducibility. KnowEdit and LEME assess specific fact changes, while CounterFact and CounterFactPlus test adaptability to counterfactuals, overlooking chained reasoning effects. MQuAKE (Zhong et al. (2023)) incorporates multi-hop query answering. Though valuable for probing broader knowledge, it does not test sequential recovery of facts, which is how users may naturally query LLMs. These evaluations primarily evaluate direct fact accuracy and multi-hop reasoning, overlooking implicit knowledge deducibility or contextual integrity. UnKEBench evaluates unstructured knowledge editing, where knowledge is represented in complex, free-form text (Deng et al. (2025)). However, it doesn't consider sequential reasoning to uncover "edited-out" facts. While prior works address ripple effects (Cohen et al. (2024); Qin et al. (2024); Chen et al. (2025); Wang et al. (2024)), event-level consistency (Peng et al. (2024)), concept-level and instance-level consistency (Niu et al. (2025)), none evaluate whether edited facts can still be recovered through multi-step reasoning. THINKEVAL fills this gap through the deep editing setting and the IFR metric.

## 3 Preliminary

A fact $k$ is represented as a triplet $(v_1, r, v_2)$, where $v_1, v_2 \in \mathcal{V}$ are entities (subject and object, respectively) and $r \in \mathcal{T}$ is a relationship connecting them. For example, (Harry Potter, school, Hogwarts) can represent the statement "Harry Potter studied at Hogwarts". A knowledge graph $\mathcal{G} = (\mathcal{V}, \mathcal{E})$ is a directed graph where $\mathcal{E} \subseteq \mathcal{V} \times \mathcal{T} \times \mathcal{V}$ is a set of edges representing facts that connect entities in $\mathcal{V}$.

### 3.1 Graph-based reasoning

Unlike rule-based systems (Wu et al. (2024)), the relationships in $\mathcal{G}$ are not derived from a predefined set of logical rules. $\mathcal{G}$ is designed such that traversing its edges naturally implies relationships between entities. We do not assume transitive closure over $\mathcal{G}$; instead, each path in $\mathcal{G}$ is retained only when it plausibly implies a meaningful relationship between the base subject and object. A path $p = (v_1, r_1, u_1, r_2, u_2, \ldots, u_{n-1}, r_n, v_2)$ in $\mathcal{G}$ with $u_i \in \mathcal{V}$ and $r_i \in \mathcal{T}$ captures a chained relationship $r$ between the endpoints, even if $(v_1, r, v_2) \notin \mathcal{E}$. For example, path (Harry Potter, house, Gryffindor, belongs to, Hogwarts) implies Harry Potter studied at Hogwarts.

### 3.2 Knowledge graphs for model editing

After editing an LLM, the internal knowledge graph $\mathcal{G} = (\mathcal{V}, \mathcal{E})$ is updated with a set of changes $\Delta\mathcal{G}$. The updated graph $\mathcal{G}' = (\mathcal{V}', \mathcal{E}')$ incorporates:

$$\textbf{Adding new edges:} \quad \Delta\mathcal{G}_{\mathrm{a}} = \{(v_1, r, v_2) \mid (v_1, r, v_2) \notin \mathcal{E}\} \tag{1}$$

$$\textbf{Removing existing edges:} \quad \Delta\mathcal{G}_{\mathrm{r}} = \{(v_1, r, v_2) \mid (v_1, r, v_2) \in \mathcal{E}\} \tag{2}$$

$$\textbf{Modifying edges:} \quad \Delta\mathcal{G}_{\mathrm{m}} = \{(v_1, r_{\mathrm{new}}, v_2) \mid (v_1, r, v_2) \in \mathcal{E}\} \tag{3}$$

The updated edge set is:

$$\mathcal{E}' = (\mathcal{E} \cup \Delta\mathcal{G}_{\mathrm{a}} \cup \Delta\mathcal{G}_{\mathrm{m}}) \setminus \Delta\mathcal{G}_{\mathrm{r}} \tag{4}$$

and

$$\mathcal{V}' = \mathcal{V} \cup \{v \mid v \in (v_1, r, v_2) \in \Delta\mathcal{G}_{\mathrm{a}} \cup \Delta\mathcal{G}_{\mathrm{m}}\} \tag{5}$$

For this study, we limit our experimentation to analyze modifying edges ($\Delta\mathcal{G}_{\mathrm{m}}$) in LLMs.

**Definition 1 Deductive Closure**: Given a knowledge graph $\mathcal{G} = (\mathcal{V}, \mathcal{E})$ with $\mathcal{E} \subseteq \mathcal{V} \times \mathcal{T} \times \mathcal{V}$ as the set of triplets, the deductive closure of $\mathcal{G}$, denoted $\Omega(\mathcal{G})$, is the set of all relationships implied by $\mathcal{G}$ such that:

1. $\mathcal{E} \subseteq \Omega(\mathcal{G})$, i.e., every triplet $(v_1, r, v_2) \in \mathcal{E}$ is a direct relationship $r$ between $v_1$ and $v_2$;

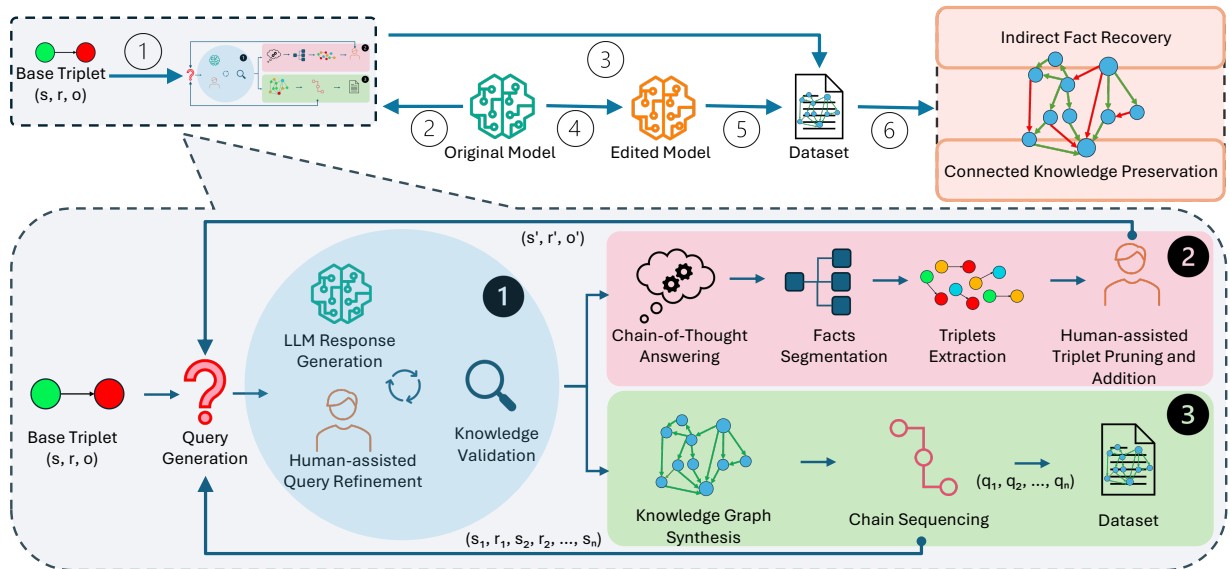

Figure 2: THINKEVAL framework. Initially, ① a base triplet and ② an LLM are utilized to ③ generate a tailored dataset reflective of the LLM's internal knowledge structure. Next, ④ the LLM is edited using an editing technique. The ⑤ the edited model is evaluated over the constructed dataset, yielding insights into the effectiveness of the editing process from deep editing perspective, ⑥ measured via IFR and Preservation.

2. $\Omega(\mathcal{G})$ is deductively closed with respect to path traversal, i.e., for every pair of entities $v_1, v_n \in \mathcal{V}$, if there exists a path $p = (v_1, r_1, v_2, r_2, \ldots, r_{n-1}, v_n)$ in $\mathcal{G}$ where each $(v_i, r_i, v_{i+1}) \in \mathcal{E}$, then $\Omega(\mathcal{G})$ includes the implied relationship $(v_1, r_p, v_n)$, where $r_p$ is the composite relationship inferred from the sequence of relationships $(r_1, r_2, \ldots, r_{n-1})$.

$\Omega(\mathcal{G})$ captures all explicit and implicit relationships derivable from $\mathcal{G}$ through its paths.

## 4 ThinkEval

We introduce THINKEVAL, a framework designed to assist users by constructing specialised knowledge graphs to quantitatively and systematically evaluate model-editing techniques via sequential prompting, as illustrated in Fig. 2. THINKEVAL operates through three interwoven components to create the dataset: ❶ Query Validation and Refinement, ❷ Automated Triplet Generation, and ❸ Graph Synthesis and Chain Sequencing. These components function in a cyclical manner, iteratively refining the knowledge representation. All the respective prompt templates are provided in Appendix E. Additionally, THINKEVAL serves as a dataset augmentation tool, capable of enhancing existing datasets by adding structured implication chains or triplets for a more comprehensive evaluation. Algorithms 1, 2 and 3 showcase the key processes utilised in THINKEVAL. For users, choosing the right model-editing technique is vital for preventing knowledge leakage and ripple effects, thereby safeguarding reliability across diverse applications.

### 4.1 Query validation and refinement

The Query Validation and Refinement component initiates knowledge extraction by validating a triplet $(s, r, o)$ or a chain $(s_1, r_1, s_2, r_2, \ldots, s_n)$, evaluating if the LLM "recognizes" the relationship. The process begins by generating targeted queries from a triplet or a chain. THINKEVAL synthesizes a knowledge graph beginning with a single triplet, and employs two prompt types:

**Triplet-based Query Generation prompt.** Targets a single triplet to elicit a direct response. For example, (`Harry Potter, student, Hogwarts`), forms the query `"Where did Harry Potter study?"`.

**Chain-based Query Generation prompt.** Probes multi-hop relationships in a chain, targeting the terminal entity, For instance, from (`Harry Potter, subject, Transfiguration, taught by, Minerva McGonagall`), `"Who teaches Transfiguration to Harry Potter?"` is crafted.

The LLM generates a response to the query, which is analyzed to determine the expected object's presence, accounting for phrasing variations or synonyms. For triplets, if the object aligns, the triplet and query are passed to the second and third components; for chains, they proceed only to the second component. If validation fails, human expertise refines the query or triplet, re-entering the cycle. After several failed cycles, indicating the model's lack of knowledge, the triplet is discarded.

## 4.2 Automated triplet generation

The Automated Triplet Generation component extracts further knowledge from the LLM using its reasoning capabilities, structuring the output into new triplets. These are cycled back to the previous component for validation. This component's functionality is demonstrated with an illustrative example in Appendix F.

The validated query is submitted to the LLM with a CoT prompt, prompting step-by-step reasoning to reveal intermediate associations alongside the final answer. The multi-sentence CoT response is segmented into discrete facts, parsing it into individual statements for triplet extraction. This segmentation simplifies the response into manageable units to preserve extracted information's integrity.

Each fact is processed by the LLM with a fact extraction prompt to generate atomic triplets, like (`Harry Potter, school, Hogwarts`) or (`Transfiguration, taught by, Minerva McGonagall`). To handle errors from hallucination or context misinterpretation, human experts prune triplets, verifying them against known facts. Authentic triplets are cycled back to the prior component, while human expertise also adds missing triplets to account for the facts overlooked by the LLM.

## 4.3 Graph synthesis and chain sequencing

The Graph Synthesis and Chain Sequencing component integrates validated triplets into a knowledge graph and structures sequences of query for inferring relationships for the dataset. The constructed chain of triplets $(s_1, r_1, s_2, r_2, \ldots, s_n)$ is cycled back to the first component.

The validated triplets are organized into the directed knowledge graph $\mathcal{G}$. $\mathcal{G}$ evolves with each iteration, incorporating new triplets, reflecting the growing complexity of the extracted knowledge.

$\mathcal{G}$ is processed to form $n$-step chains $(s_1, r_1, s_2, r_2, \ldots, s_n)$, to serve two purposes: (1) chains are sent to first component for multi-hop query generation, and (2) their query sequences are compiled into a dataset. These are capped at five steps, as longer chains tend to diminish the impact of associating the subject with the object. The effectiveness of extracting meaningful relationships decreases with increasing number of queries, making shorter chains more impactful for establishing clear associations.

# 5 Deep Editing

Existing model editing techniques often modify a target fact in isolation, often overlooking logically connected facts and thereby introducing inconsistencies. We introduce **deep editing** as a new evaluation setting that characterizes whether an edit has been performed consistently with respect to related knowledge. Unlike event-level editing (Deng et al. (2025)), which focuses on updating facts tied to a specific event, deep editing evaluates whether edits propagate appropriately through a network of logically connected facts, while preserving unrelated knowledge. A fact is deep edited up to $n$-steps if it cannot be empirically deduced from the retained knowledge through any sequence of reasoning steps of length $\leq n$, where $n$ is a user-defined parameter reflecting computational or application constraints. We provide a comparison of deep editing with other model-editing settings in Appendix C.

Suppose an editing technique $\mathcal{A}$ modifies a target fact $t = (v_1, r, v_2) \in \mathcal{E}$, resulting in modified triplets $\mathcal{E}_t^{\mathcal{A}} \subseteq \mathcal{E}$ to form an updated graph $\mathcal{G}' = (\mathcal{V}', \mathcal{E}')$. Here, $\mathcal{E}' = (\mathcal{E} \setminus \mathcal{E}_t^{\mathcal{A}}) \cup \Delta\mathcal{E}_t^{\mathcal{A}}$, where $\Delta\mathcal{E}_t^{\mathcal{A}} = \{(v_i, r_i', v_j) \mid$

---

**Algorithm 1** Query Validation and Refinement

---

1: **Input:** Knowledge input $\mathcal{I}$ (either triplet $t = (s, r, o)$ or chain $c = (s_1, r_1, s_2, r_2, \ldots, s_n)$), LLM $\mathcal{M}$, max iterations $k_{\max}$
2: **Output:** Validated triplet $t'$ or chain $c'$, or $\emptyset$ (discarded)
3: $q \leftarrow \text{GenerateQuery}(\mathcal{I})$               $\triangleright$ Triplet or chain-based prompt
4: $k \leftarrow 0$
5: **while** $k < k_{\max}$ **do**
6:      $\mathcal{R} \leftarrow \mathcal{M}(q)$                    $\triangleright$ LLM response
7:      **if** $o \in \mathcal{R}$ **then**             $\triangleright$ Object present in response
8:          **if** $\mathcal{I}$ is chain $c$ **then**
9:              $c' \leftarrow c$
10:              **cycle** $q$ to Algorithm 3
11:          **else**
12:              $t' \leftarrow t$
13:          **end if**
14:          **return** $q$ to Algorithm 2
15:      **else**
16:          $I, q \leftarrow \text{HumanRefine}(q, \mathcal{R}, \mathcal{I})$     $\triangleright$ Refine based on input type
17:          $k \leftarrow k + 1$
18:      **end if**
19: **end while**
20: **discard** $t$                   $\triangleright$ No validation after $k_{\max}$
21: **return** $\emptyset$

---

**Algorithm 2** Automated Triplet Generation

---

1: **Input:** Validated query $q$, LLM $\mathcal{M}$
2: **Output:** Set of new triplets $\mathcal{T}_{\text{new}}$
3: $\mathcal{R} \leftarrow \mathcal{M}(q, \text{CoT})$              $\triangleright$ CoT response
4: $\mathcal{F} \leftarrow \text{SegmentFacts}(\mathcal{R})$           $\triangleright$ Set of facts
5: $\mathcal{T}_{\text{temp}} \leftarrow \emptyset$
6: **for** $f \in \mathcal{F}$ **do**
7:      $\mathcal{T}_f \leftarrow \mathcal{M}(f, \text{ExtractPrompt})$      $\triangleright$ Extract triplets
8:      $\mathcal{T}_{\text{temp}} \leftarrow \mathcal{T}_{\text{temp}} \cup \mathcal{T}_f$
9: **end for**
10: $\mathcal{T}_{\text{new}} \leftarrow \text{HumanPruneAndAdd}(\mathcal{T}_{\text{temp}})$     $\triangleright$ Prune and augment
11: **return** $\mathcal{T}_{\text{new}}$ to Algorithm 1

---

**Algorithm 3** Graph Synthesis and Chain Sequencing

---

1: **Input:** Validated triplet $t = (s, r, o)$, corresponding validated query $q$
2: **Output:** Knowledge graph $\mathcal{G}$, dataset $\mathcal{D}$
3: $\mathcal{V} \leftarrow \mathcal{V} \cup \{s, o\}$
4: $\mathcal{E} \leftarrow \mathcal{E} \cup \{(s, r, o)\}$
5: $\mathcal{G} \leftarrow (\mathcal{V}, \mathcal{E})$                 $\triangleright$ Initialize graph
6: $\mathcal{C} \leftarrow \text{GenerateChains}(\mathcal{G})$         $\triangleright$ Chains from $\mathcal{G}$
7: $\mathcal{D} \leftarrow \emptyset$                   $\triangleright$ Initialize dataset
8: **for** $c = (s_1, r_1, s_2, r_2, \ldots, s_n) \in \mathcal{C}$ **do**
9:      $\mathcal{Q}_c \leftarrow \text{GetQuerySequence}(c, l_{\max})$     $\triangleright$ Query sequence with maximum length as $l_{\max}$
10:      $\mathcal{D} \leftarrow \mathcal{D} \cup \{\mathcal{Q}_c\}$
11:      **cycle** $c$ to Algorithm 1
12: **end for**
13: **return** $\mathcal{G}, \mathcal{D}$

---

$(v_i, r_i, v_j) \in \mathcal{E}^{\mathcal{A}}{}_t\}$ represents the modified triplets in $\mathcal{E}^{\mathcal{A}}{}_t$, with the new relationships $r'_i \in \mathcal{T}$. If technique $\mathcal{A}$ deep edits the fact $t$, then $t$ must not be implied by any path in $\mathcal{G}'$, i.e., $t$ should not belong to $\Omega(\mathcal{G}')$.

**Definition 2 Deep Editing**: An editing technique $\mathcal{A}$ deep edits the fact $t = (v_1, r, v_2)$ with respect to the knowledge graph $\mathcal{G} = (\mathcal{V}, \mathcal{E})$ if $t$ does not belong to the deductive closure $\Omega(\mathcal{G}')$, where $\mathcal{G}' = (\mathcal{V}', \mathcal{E}')$ is the updated graph, and $\mathcal{E}' = (\mathcal{E} \setminus \mathcal{E}^{\mathcal{A}}{}_t) \cup \Delta\mathcal{E}^{\mathcal{A}}{}_t$ reflects the modified triplets. Formally, $(v_1, r, v_2) \notin \Omega(\mathcal{G}')$, i.e., no path in $\mathcal{G}'$ implies the original relationship $r$ between $v_1$ and $v_2$.

## 5.1 Evaluation metrics for deep editing

We outline the Counterfact (Meng et al. (2022)) metrics to evaluate model editing in Appendix D. For deep editing evaluation, we propose a new evaluation metric Indirect Fact Recovery (IFR).

IFR measures the extent to which the original fact remains deducible despite the edit. It is implied by paths in $\mathcal{G} = (\mathcal{V}, \mathcal{E})$ after editing to $\mathcal{G}' = (\mathcal{V}', \mathcal{E}')$. Let $\mathcal{D} = \{\mathcal{S}_1, \ldots, \mathcal{S}_m\}$ be sequences of queries corresponding to chains from $\mathcal{G}$, where $\mathcal{S}_i = \{q_{i1}, \ldots, q_{in_i}\}$ has length $n_i$, and each $q_{ij}$ probes a step in a path implying a target fact. Let $\mathcal{P}_{\mathcal{G}} = \{\mathcal{P}_{\mathcal{S}_1}, \ldots, \mathcal{P}_{\mathcal{S}_m}\}$ and $\mathcal{P}_{\mathcal{G}'} = \{\mathcal{P}'_{\mathcal{S}_1}, \ldots, \mathcal{P}'_{\mathcal{S}_m}\}$, where $\mathcal{P}_{\mathcal{S}_i} = \{p_{i1}, \ldots, p_{in_i}\}$ and $\mathcal{P}'_{\mathcal{S}_i} = \{p'_{i1}, \ldots, p'_{in_i}\}$ are the probabilities of the original target output occurring in the model response to each $q_{ij}$ in $\mathcal{G}$ and $\mathcal{G}'$, respectively.

**Definition 3 Indirect Fact Recovery**:

$$\text{IFR}(\mathcal{G}, \mathcal{G}', \mathcal{D}, P_{\mathcal{G}}, P_{\mathcal{G}'}) = \begin{cases} \dfrac{\displaystyle\sum_{\mathcal{S}_i \in \mathcal{D}'} \dfrac{\mathcal{R}'_{\mathcal{S}_i}/\mathcal{R}_{\mathcal{S}_i}}{\sqrt{n_i}}}{\displaystyle\sum_{\mathcal{S}_i \in \mathcal{D}'} \dfrac{1}{\sqrt{n_i}}}, & \text{if } \mathcal{D}' \neq \emptyset, \\[1em] 0, & \text{otherwise,} \end{cases} \tag{6}$$

where $\mathcal{R}_{\mathcal{S}_i} = \prod_{j=1}^{n_i} p_{ij}$, $\mathcal{R}'_{\mathcal{S}_i} = \prod_{j=1}^{n_i} p'_{ij}$, and $\mathcal{D}' = \{\mathcal{S}_i \in \mathcal{D} \mid \mathcal{R}_{\mathcal{S}_i} \neq 0\}$.

IFR computes the weighted average of retention ratios $\mathcal{R}'_{\mathcal{S}_i}/\mathcal{R}_{\mathcal{S}_i}$, normalized by $\sqrt{n_i}$, across all the sequences to emphasize the higher implication strength of the shorter paths. Here, $p_{ij}$ and $p'_{ij}$ reflect probability of the original target output (object in a triplet) in the model's response to $q_{ij}$ before and after editing. A high IFR indicates significant deducibility of the original target fact, suggesting incomplete deep editing. A low IFR reflects that original fact is hardly deducible after editing.

We use Preservation (Cohen et al. (2024)) to evaluate the the broader contextual knowledge integrity in deep editing. While RippleEdits utilises it to evaluate other subject-relationships, we utilise it to measure retention of accuracy across facts in $\mathcal{G} = (\mathcal{V}, \mathcal{E})$ after editing to $\mathcal{G}' = (\mathcal{V}', \mathcal{E}')$, excluding those which are not a part of the broader contextual knowledge (links directly linked to the original subject unless specified otherwise). Let $t_0 = (s_0, r_0, o_0)$ be the original triplet, and $\mathcal{E}_{\text{ind}} = \{(s, r, o) \in \mathcal{E} \mid s \neq s_0\}$ as the set of triplets where the subject is not $s_0$. For each $t = (s, r, o) \in \mathcal{E}_{\text{ind}}$, let $p_t \in [0, 1]$ and $p'_t \in [0, 1]$ be probabilities of the original target output $o$ occurring in the model's response to a query $q_t$ in $\mathcal{G}$ and $\mathcal{G}'$, respectively.

**Definition 4 Preservation**:

$$\text{Preservation}(\mathcal{G}, \mathcal{G}', \mathcal{E}_{\text{ind}}) = \begin{cases} \dfrac{1}{|\mathcal{E}'_{\text{ind}}|} \displaystyle\sum_{t \in \mathcal{E}'_{\text{ind}}} \dfrac{p'_t}{p_t}, & \text{if } \mathcal{E}'_{\text{ind}} \neq \emptyset, \\[1em] 1, & \text{otherwise,} \end{cases} \tag{7}$$

where $\mathcal{E}'_{\text{ind}} = \{t \in \mathcal{E}_{\text{ind}} \mid p_t \neq 0\}$.

Preservation averages the ratios $p'_t/p_t$ over all triplets in $\mathcal{E}'_{\text{ind}}$, where $p_t$ and $p'_t$ represent probabilities of the original object $o$ in response to $q_t$ before and after editing. A high Preservation indicates strong preservation of broader contextual knowledge, suggesting that the editing process successfully avoided catastrophic

forgetting over them. A low Preservation reflects a loss of accuracy in these facts, implying unintended knowledge degradation within the model.

## 6 Performance Evaluation

Using the KnowGIC benchmark, we evaluated five model-editing techniques: AlphaEdit, RECT, ROME, MEMIT and PRUNE via IFR, Preservation and Efficacy on three models, Qwen2.5 (7B), Llama3 (8B) and GPT2-XL (1.5B). All implementations are sourced from Jiang (2024). All experiments are performed on an NVIDIA A100 80GB GPU.

### 6.1 KnowGIC Benchmark

To evaluate in the deep editing setting, we introduce the KnowGIC benchmark, which is built using THINKEVAL. Extending the multi-hop reasoning datasets, KnowGIC consists of $n$-step implication chains, or sequences of n prompts to probe specific relationships via multiple reasoning steps. These chains capture direct and implied relationships for a fine-grained evaluation. Unlike MQuAKE's n-hop queries, which encapsulate multiple reasoning hops in a single query, each implication chain step probes a single hop in the knowledge graph. If the original fact can be logically reconstructed through such sequential prompting, it signals incomplete editing. Conversely, breaks in the chain that disrupt unrelated but connected facts reveal unintended ripple effects. Table 1 presents the distribution of chain lengths in the dataset. All chains have been manually reviewed to remove redundancy, factual inaccuracies, and irrelevant triplets, ensuring high quality and relevance for deep editing evaluation.

Table 1: Distribution of multi-step implication chains in the KnowGIC benchmark.

| Chain length (steps) | 1 | 2 | 3 | 4 | 5 |
|---|---|---|---|---|---|
| Number of Chains | 24 | 108 | 227 | 428 | 619 |

We selected various base templates from the MQuAKE dataset across diverse relational domains, generating knowledge graphs using THINKEVAL for their samples. To illustrate the depth of connected facts and how interlinked facts can help extract the original relationship, we include an additional sample (`Harry Potter, school, Hogwarts`). Using THINKEVAL, we grow its knowledge graph till it achieves 100 implication chains from `Harry Potter` to `Hogwarts`. The KnowGIC benchmark, including this sample, comprises 1,406 $n$-step reasoning paths. Extensive details of the dataset are in Appendix G.

### 6.2 Motivation for new metrics

Existing metrics like Efficacy and Specificity provide valuable insights into different aspects of model editing, but are insufficient for deep editing evaluation. Efficacy focuses on immediate effect of the edit, overlooking whether the original fact can be inferred via sequential reasoning. Specificity focuses on the correctness of the LLM's response in terms of higher probability assigned to the correct response. Similarly, Consistency focuses only on factual coherence. Deep editing quantifies how much the original fact can be deduced even via sequential reasoning. Hence, a new metric is essential to evaluate this dimension comprehensively.

Table 2: `Harry Potter` case-study setup and results.

| | |
|---|---|
| **Relationship triplet** | (Harry Potter, school, Hogwarts) |
| **Edit request** | Hogwarts → Ilvermorny |
| **Model** | Llama3-8B-Instruct |
| **Edit technique** | AlphaEdit |
| **Paths identified** | 100 |
| **Active paths post edit** | 80 |
| **Efficacy** (↑) | 1.000 |
| **IFR** (↓) | 0.780 |

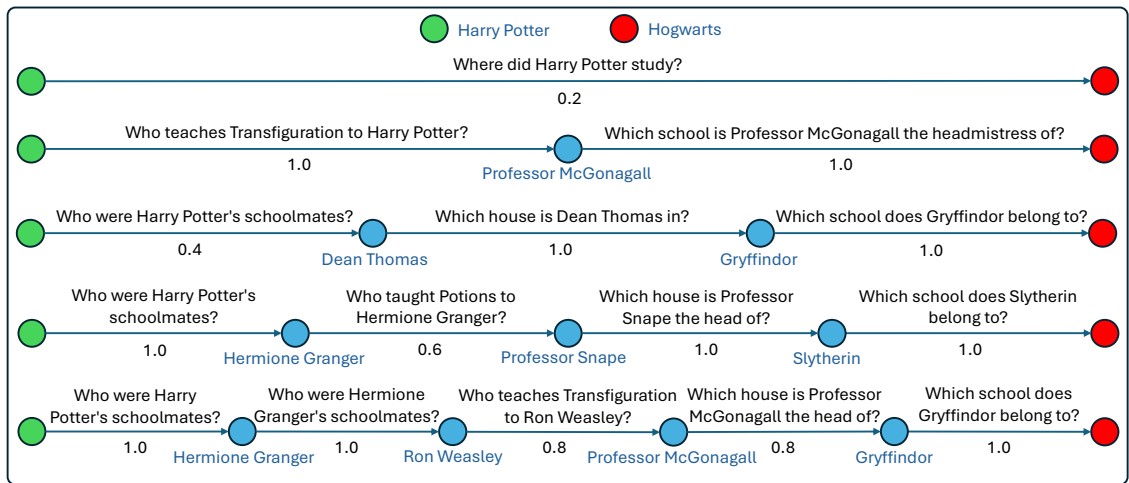

Figure 3: Samples of $n$-step chains from `Harry Potter` case-study, leading to original fact leakage even after editing. The number below each link represents the ratio of responses (out of five generations) that retain the original output, quantifying the extent to which the LLM reveals the initial fact via indirect reasoning.

We conduct a case-study on the sample (`Harry Potter, school, Hogwarts`) from KnowGIC to evaluate the need for new metrics in deep editing, aiming to edit the fact (`Hogwarts → Ilvermorny`). The setup is detailed in Table 2. For each query in the implication chains, we generate five distinct responses from both the original and edited models. An Efficacy of 1.000 shows that the LLM responds with `Ilvermorny` when queried directly. However, it does not account for original fact recovery through indirect reasoning paths. An overall IFR of 0.780 suggests that the original fact still remains deducible to a significant extent. Figure 3 shows samples from these 80 active chains.

Table 3: $n$-step IFR for the `Harry Potter` case-study.

| n | $n$-step chains | Active chains post edit | $n$-step IFR |
|---|---|---|---|
| 1 | 1 | 1 | 0.200 |
| 2 | 20 | 14 | 0.814 |
| 3 | 30 | 23 | 0.736 |
| 4 | 35 | 30 | 0.484 |
| 5 | 14 | 12 | 0.402 |

As shown in Table 3, the 1-step chain with IFR of 0.200 shows effective suppression of direct inference. However, the 2-step and 3-step chains exhibit high IFR values of 0.814 and 0.736 respectively, demonstrating high indirect fact leakage through short reasoning paths. The 4-step and 5-step chains show moderate IFR values of 0.484 and 0.402, further demonstrating persistent leakage across longer paths. Despite an Efficacy of 1.00, the original fact persists through multi-step reasoning, indicating AlphaEdit's surface-level success but failure in deeper knowledge updates. For an editing method to qualify as successful under the deep editing paradigm, the original fact must be eliminated across all reasoning paths. This case-study also highlights that traditional metrics like Efficacy are insufficient for deep editing. IFR's focus on multi-step chains provides a more robust evaluation. Further details of the case-study are in Appendix H.

## 6.3 Results and discussions

Table 4 presents the overall IFR, Preservation and Efficacy values for different editing techniques across GPT2-XL, Llama3-8B, and Qwen2.5-7B. For Llama3-8B, AlphaEdit achieves the highest Efficacy of 0.958, followed by PRUNE at 0.875, indicating strong performance in updating target facts. For Qwen2.5-7B, AlphaEdit achieves a perfect Efficacy of 1.000, with MEMIT also performing strongly at 0.958. For GPT2-

Table 4: Comparison of model-editing techniques for various models using IFR, Preservation (Pres.), and Efficacy (Eff.). Values are color-coded from red (lower performance) to green (higher performance).

| Model | Technique | IFR (↓) | | | | | | Pres. (↑) | Eff. (↑) |
|---|---|---|---|---|---|---|---|---|---|
| | | Overall | 1-step | 2-step | 3-step | 4-step | 5-step | | |
| Llama3-8B | AlphaEdit | 0.509 | 0.413 | 0.557 | 0.641 | 0.626 | 0.351 | **0.890** | **0.958** |
| | MEMIT | 0.550 | 0.434 | 0.628 | 0.634 | 0.599 | 0.461 | 0.857 | 0.792 |
| | RECT | 0.487 | 0.454 | 0.465 | 0.552 | 0.564 | 0.406 | 0.823 | 0.708 |
| | PRUNE | 0.187 | 0.178 | 0.187 | 0.195 | 0.195 | 0.178 | 0.696 | 0.875 |
| | ROME | **0.134** | 0.173 | 0.116 | 0.222 | 0.177 | 0.060 | 0.685 | 0.833 |
| Qwen2.5-7B | AlphaEdit | 0.496 | 0.108 | 0.269 | 0.563 | 0.579 | 0.497 | **0.869** | **1.000** |
| | MEMIT | 0.577 | 0.097 | 0.287 | 0.535 | 0.549 | 0.741 | 0.859 | 0.958 |
| | RECT | 0.569 | 0.304 | 0.561 | 0.599 | 0.679 | 0.494 | 0.851 | 0.750 |
| | PRUNE | 0.514 | 0.378 | 0.427 | 0.521 | 0.496 | 0.561 | 0.765 | 0.792 |
| | ROME | **0.332** | 0.177 | 0.212 | 0.173 | 0.381 | 0.417 | 0.669 | 0.708 |
| GPT2-XL | AlphaEdit | 0.461 | 0.199 | 0.547 | 0.639 | 0.565 | 0.294 | 0.663 | **1.000** |
| | MEMIT | 0.484 | 0.271 | 0.518 | 0.654 | 0.549 | 0.362 | 0.683 | 0.875 |
| | RECT | 0.516 | 0.482 | 0.539 | 0.673 | 0.548 | 0.413 | **0.689** | 0.708 |
| | PRUNE | 0.211 | 0.238 | 0.208 | 0.209 | 0.219 | 0.203 | 0.587 | 0.917 |
| | ROME | **0.196** | 0.299 | 0.237 | 0.263 | 0.278 | 0.158 | 0.500 | 0.792 |

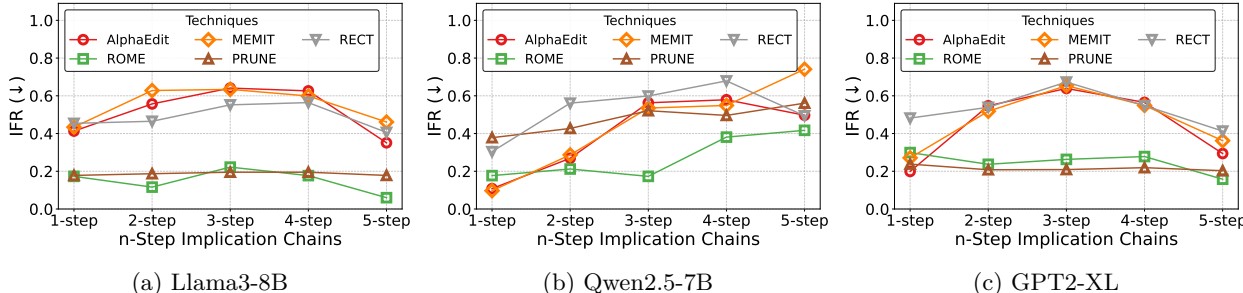

(a) Llama3-8B      (b) Qwen2.5-7B      (c) GPT2-XL

Figure 4: IFR for *n*-step samples. A lower IFR implies lower deducibility of the supposedly-edited fact.

XL, both AlphaEdit and PRUNE achieve high Efficacy scores of 1.000 and 0.917, respectively. Efficacy, however, provides limited insights into the performance of these techniques by focusing on direct edit success while neglecting the broader implications of these edits.

In contrast, IFR scores reveal significant variability across models, as shown in Fig. 4. For Llama3-8B, AlphaEdit exhibits an IFR increase from 0.413 (1-step) to 0.641 (3-step) before declining to 0.351 (5-step), indicating that the original fact remains highly deducible, particularly via mid-range chains (3-step). On Qwen2.5-7B, AlphaEdit's IFR rises steadily from 0.108 (1-step) to 0.497 (5-step), showing persistent fact leakage. For GPT2-XL, AlphaEdit's IFR increases from 0.199 (1-step) to 0.639 (3-step) before dropping to 0.294 (5-step), following a similar rise-then-fall pattern. MEMIT on Qwen2.5-7B shows a sharp IFR increase from 0.097 (1-step) to 0.741 (5-step), while on Llama3-8B, it follows a comparable rise-then-fall trend (0.434 at 1-step, 0.634 at 3-step, 0.461 at 5-step). For GPT2-XL, MEMIT's IFR rises from 0.271 (1-step) to 0.654 (3-step) before declining to 0.362 (5-step), highlighting contrasting leakage patterns. ROME and PRUNE maintain low IFR across all models, with ROME on Llama3-8B dropping to 0.060 at 5-step, and on GPT2-XL to 0.158, indicating strong reduction in original fact leakage. However, this may suggest ripple effects in related knowledge. These techniques are typically designed to edit direct facts (1-step chains), expecting their impact to diminish as chain length increases, ideally resulting in an increasing IFR curve. Most techniques on Qwen2.5-7B align with this expectation, but deviate on other models, showing a peak before declining.

The rise-then-fall IFR pattern in AlphaEdit, MEMIT, and RECT on Llama3-8B and GPT2-XL is attributed to the interplay between their editing strategies and ripple effects. As IFR relies on the product of link strengths in an implication chain, longer chains (4-5 steps) are more likely to encounter weakened links, reducing deducibility due to eroded related knowledge. These techniques disrupt broader contextual connections, causing a decline in IFR beyond mid-range chains. In contrast, ROME and PRUNE show lower

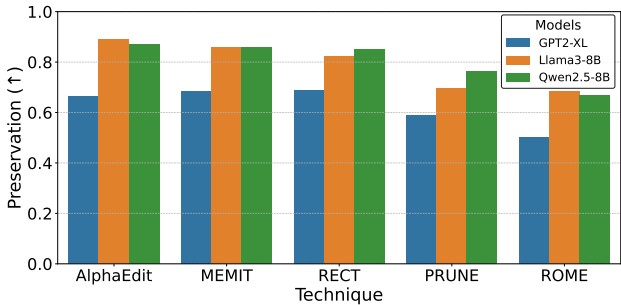

Figure 5: Preservation for different model-editing techniques. A higher Preservation indicates stronger retention of broader context integrity.

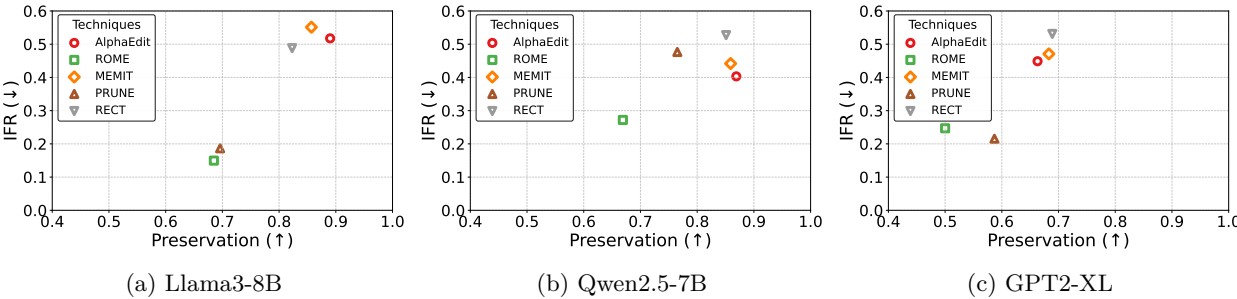

(a) Llama3-8B      (b) Qwen2.5-7B      (c) GPT2-XL

Figure 6: IFR vs. Preservation for different model-editing techniques, illustrating trade-offs in indirect fact recovery and related knowledge preservation.

IFR across all models, risking over-editing, as reflected by their Preservation scores. As illustrated in Fig. 5, ROME's Preservation is relatively low (0.685 on Llama3-8B, 0.669 on Qwen2.5-7B, 0.500 on GPT2-XL), indicating poor preservation of related facts, while AlphaEdit achieves higher Preservation (0.890 on Llama3-8B, 0.869 on Qwen2.5-7B, 0.663 on GPT2-XL). GPT2-XL models show low preservation across all techniques due to high entanglement of facts stemming from its smaller size (Qin et al. (2024)). For Qwen2.5-7B, the predominantly increasing IFR curves, such as AlphaEdit's rise from 0.108 (1-step) to 0.497 (5-step), prompt additional investigations to explore its robustness over extended implication chains, detailed in Appendix A. Fig. 6 highlights the varying performance of these techniques. For users, IFR delivers critical insights into the subtle impacts of edits across implication chains, revealing strengths and weaknesses missed by existing metrics and underscoring the importance of systematic evaluation frameworks such as THINKEVAL.

## 7 Conclusion

In this study, we discuss the challenge of maintaining consistent knowledge in LLMs under model editing. We introduced THINKEVAL, a framework which offers users involved in updating and deploying a model a systematic evaluation of editing techniques. Within this framework, we defined the deep editing evaluation setting to quantify the original knowledge leakage and ripple effects caused by edits, and proposed Indirect Fact Recovery (IFR) as a metric to quantify such leakage, thus assisting these users in choosing the suitable model and the right editing technique for their use case. For evaluation, we constructed the KnowGIC benchmark, a dataset of multi-step reasoning paths. Our experiments expose significant limitations in five state-of-the-art editing techniques across three models. While these techniques effectively deal with directly queried facts, they frequently fail to eliminate indirect knowledge leakage, allowing the original information to persist. Additionally, edits often introduce unintended ripple effects that disrupt the coherence of surrounding contextual knowledge. These findings highlight the need for evaluation frameworks like THINKEVAL to guide users in selecting editing methods and for the development of more holistic techniques that update knowledge reliably without compromising contextual integrity.

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

**Target Triplet: (Hermione Granger, taught Ancient Runes by, Bathsheda Babbling)**

**Prompt**: Who teaches Ancient Runes to Hermione Granger?
**Response**: In the Harry Potter series, Ancient Runes is not actually a course taught at Hogwarts School of Witchcraft and Wizardry. Instead, Hermione Granger learns about ancient languages and scripts on her own. She shows an interest in Ancient Runes and spends time studying them independently.
The subject that comes closest to Ancient Runes within the wizarding curriculum at Hogwarts is Muggle Studies, which deals with the study of non-magical cultures and their inventions, including writing systems and historical texts. However, this is not directly related to Hermione's interest in ancient languages from her own studies.
In the books, Hermione's knowledge of ancient scripts such as Old Norse Runes and other similar scripts comes from her independent reading and research outside of class.

**Prompt**: What does  Bathsheda Babbling teach to Hermione Granger?
**Response**: It seems there might be a mix-up in the names or titles. There is no known character named "Bathsheda Babbling" in the Harry Potter series written by J.K. Rowling, nor do they teach anyone directly.
However, I can help clarify if you meant "Bathilda Bagshot," who appears in some of the supplementary materials and books related to the Harry Potter universe. Bathilda Bagshot is a character known for her magical history books, but she doesn't specifically teach Hermione Granger anything in the main series.
If you're referring to a specific lesson or teaching that Hermione received from another character (like a professor), could you please provide more details? This would help me give you a more accurate and detailed answer.

**Prompt**: Did Bathsheda Babbling teach Hermione Granger Ancient Runes?
**Response**: No, Bathsheda Babbling did not teach Hermione Granger Ancient Runes. The character you might be thinking of is Bathilda Bagshot, who wrote "A History of Magic," a book that  Hermione used to learn about Ancient Runes. However, Hermione's primary teacher for Ancient Runes was Professor Barty Crouch Jr., in his guise as Mad-Eye Moody during her  third year at Hogwarts.
Hermione studied Ancient Runes alongside other subjects and delved into the subject with great interest due to its magical and historical significance. She often relied on books like "A History of Magic" for additional reading and knowledge, but her main instruction came from Professor Crouch/Moody in class.

Figure 7: Qwen2.5-7B's Responses to various prompts for the target triplet (Hermione Granger, taught Ancient Runes by, Bathsheda Babbling)

# A    Original Model Performance on KnowGIC Queries

Qwen2.5-7B accurately answered only 81% of the queries included across all implication chains, causing inference from a significant number of chains infeasible. This may be due to the fact that Llama3-8B was utilised for fact extraction for the KnowGIC dataset. To explore this further, we conduct additional tests using Ollama (Ollama Team (2023)). These tests are run on a sample of the queries which failed to answer correctly while inferencing with the original unedited model. This is to determine whether existing queries in the dataset are not able to fetch the expected responses, or the knowledge itself is not there in the model.

Fig 7 illustrates Qwen2.5-7B's responses to three prompts designed to test its knowledge of the triplet (Hermione Granger, taught Ancient Runes by, Bathsheda Babbling). These include a direct query where the object is not present within the prompt ("Who teaches Ancient Runes to Hermione Granger?"), a query where the relationship is not present within the prompt ("What does Bathsheda Babbling teach to Hermione Granger?"), and a prompt which contains all the three elements of the triplet ("Did Bathsheda Babbling teach Hermione Granger Ancient Runes?"). The model's responses reveal significant knowledge gaps, such as denying Ancient Runes as a Hogwarts course, confusing Bathsheda Babbling with Bathilda Bagshot, and incorrectly attributing other entities. Thus, we can conclude there are gaps in Qwen2.5-7B's knowledge with respect to a few queries in KnowGIC.

Fig 8 illustrates Qwen2.5-7B's responses to three prompts designed to test its knowledge of the triplet (Nicole Richie, spouse, Joel Madden). The prompts include a direct query where the object is not present within the prompt ("Who is Nicole Richie's spouse?"), a query where the relationship is not present within the prompt ("Who is Joel Madden's spouse?"), and a prompt which contains all the three elements of the triplet ("Is Joel Madden's spouse Nicole Richie?"). The model's responses are inconsistent: it incorrectly names Blake Shelton as Nicole Richie's spouse and Ashlee Simpson as Joel Madden's spouse, despite later confirming that Joel Madden is married to Nicole Richie.

Evaluating another triplet shown in Fig 9, Don Rich was a key member of The Buckaroos and not of The Jordanaires. The Buckaroos is the backing band for Buck Owens, not Elvis Presley or Charlie Rich. Don

---

**Target Triplet: (Nicole Richie, spouse, Joel Madden)**

**Prompt**: Who is Nicole Richie's spouse?
**Response**: Nicole Richie's spouse is Blake Shelton. They got married on September 25, 2016. Blake Shelton is an American singer and musician, best known for being a coach on seasons 8-13 of "The Voice" and for his successful career in country music.

**Prompt**: Who is Joel Madden's spouse?
**Response**: Joel Madden, the lead singer of the band Good Charlotte and formerly of No Doubt (not to be confused with Tony Kanal who was in No Doubt), is married to Ashlee Simpson. They got married on April 15, 2006, and later had a daughter named Wilder Amour Madden. However, Joel and Ashlee/ divorced in 2019.

**Prompt**: Is Joel Madden's spouse Nicole Richie?
**Response**: Yes, Joel Madden is married to Nicole Richie. They got married on September 29, 2007, and have two children together: Jasmine Joy Madden (born in 2010) and Haven James Madden (born in 2014). Joel is the lead singer of the band Good Charlotte, while Nicole Richie is a well-known fashion designer, model, and television personality.

---

Figure 8: Qwen2.5-7B's Responses to various prompts for the target triplet (Nicole Richie, spouse, Joel Madden)

---

**Target Triplet: (Don Rich, band, The Buckaroos)**

**Prompt**: Which band was Don Rich a part of?
**Response**: Don Rich was a member of the country music band The Jordanaires. He joined the group in 1957 and was known for his smooth baritone vocals and his harmonies, which contributed significantly to the sound of many famous artists including Elvis Presley, Patsy Cline, and Jim Reeves. The Jordanaires are one of the most successful vocal groups in country music history, backing numerous chart-topping hits throughout their long career.

**Prompt**: Was Don Rich a member of The Buckaroos music band?
**Response**: Don Rich was indeed associated with The Buckaroos, but his role wasn't as a member of the band itself. Don Rich was a guitarist who played for the Charlie Rich Band, which supported The Buckaroos during their performances in the mid-1960s. Charlie Rich was a famous American country and rockabilly singer, and Don Rich's guitar work significantly contributed to the sound that made Charlie Rich successful. The Buckaroos were an influential band associated with Elvis Presley. They played at Presley's home studio, Graceland, and were known for their energetic performances and distinctive sound in rockabilly and country music.

---

Figure 9: Qwen2.5-7B's Responses to various prompts for the target triplet (Don Rich, band, The Buckaroos)

---

**Target Triplet: (Benjamin Keough, mother, Lisa Marie Presley)**

**Prompt**: Who is the mother of Benjamin Keough?
**Response**: I'm sorry, but I don't have any specific information about who the mother of Benjamin Keough might be. Without additional context or details, it's difficult to provide accurate information. The name "Benjamin Keough" does not correspond to a widely known historical figure or public personality that I can confidently identify as needing such information. If you could provide more context or specify which Benjamin Keough you're referring to (e.g., the time period, location, or any other relevant details), I might be able to assist you better.

**Prompt**: Is Lisa Marie Presley the mother of Benjamin Keough?
**Response**: Lisa Marie Presley is not the mother of Benjamin Keough. Benjamin Keough's biological parents are Justin and Erin Keough. Lisa Marie Presley passed away in 2023, and there
is no public information suggesting she has a child named Benjamin Keough. If you have more specific details or context about your question, feel free to provide it for
further clarification!

---

Figure 10: Qwen2.5-7B's Responses to various prompts for the target triplet (Benjamin Keough, mother, Lisa Marie Presley)

Rich wasn't just a supporting guitarist for another act; he was the lead guitarist and a core part of the band from early 1960s until 1974. The Buckaroos were not associated with Elvis Presley or Graceland. They were Buck Owens' band, formed in Bakersfield, California. Charlie Rich, a separate artist had his own career and band, but Don Rich had no notable connection to him.

Evaluating the triplet shown in Fig 10, Lisa Marie Presley is mother of Benjamin Keough. Benjamin Keough was born on October 21, 1992, to Lisa Marie Presley and her husband Danny Keough. There's no evidence of a Benjamin Keough born to a Justin and Erin Keough. Hence, for evaluating model editing, there is a need to customize the prompts which may be specific to the model itself.

IFR trends for Qwen2.5-7B differ significantly from those of Llama3-8B, which may be due to the reason mentioned above. In Qwen2.5-7B, a higher number of implication chains remain inactive, leading to deviations from the observed trends in Llama3-8B. These errors highlight the challenges in model editing, as Qwen2.5-7B fails to accurately reflect the target triplets, underscoring the need for frameworks like THINKEVAL that enable model-specific prompt customization and identification of model-specific relationships to address such deficiencies effectively.

## B   On Human Involvement and Graph Completion

Human involvement is essential in current LLM research due to the well-known issue of hallucination. Models often produce plausible but incorrect information. Several existing knowledge editing datasets also incorporate human involvement (Deng et al. (2025); Peng et al. (2024); Tsaneva et al. (2025)). Similarly, various recent LLM studies leverage HITL feedback (John et al. (2025); Kadam (2024)) to ensure factual accuracy, address data sparsity, and improve robustness. Our method follows this standard practice to prioritize precision and interpretability.

The core objective of THINKEVAL is not to obtain complete knowledge graphs but to operationalize deep editing. Because of how knowledge is ever-evolving, it is extremely difficult to empirically establish that any knowledge graph is complete. Authors in Yang et al. (2022) and Peng et al. (2023) discuss the theory of Knowledge Graph Completion. Although there are many methods for constructing knowledge graphs, it is still infeasible to create comprehensive representations of all the knowledge in a field. They further dive into the open-world problem and the incompleteness of knowledge graphs (the open-world assumption).

For the Harry Potter case-study in Section 6.2, THINKEVAL generates the knowledge graph up-to 100 chains with minimal human intervention. This is not a hard limit but a design choice for our evaluation. These graphs can be expanded further using THINKEVAL, making it a flexible and scalable tool for probing deep relational entanglements in LLMs.

## C   Comparison of Model-Editing Settings

Table 5 provides a detailed comparison between various model-editing settings. The use of specialised knowledge graphs in deep editing to infer over implication chains enables us to uncover subtle editing failures is something which other benchmarks or settings may miss. We measure whether the original facts remain deducible post-edit through reasoning paths, and also test how edits propagate through related facts. Either of the other two model-editing settings don't consider sequential reasoning to uncover "edited-out" facts.

## D   Counterfact Metrics for Model Editing

In evaluating model editing for LLMs, a variety of metrics assess different aspects of edit quality and impact. Based on prior work (Meng et al. (2022)), we outline the Counterfact evaluation metrics for assessing model editing in LLMs. Existing metrics are defined for an LLM $f_\theta$, a knowledge fact prompt $(s_i, r_i)$, an edited target output $o_i$, and the model's original output $o_c^i$, providing a comprehensive framework to evaluate the effectiveness and robustness of edits:

Table 5: Comparison of Model-Editing Settings

| Model Editing Setting | Unstructured | Event-based | Deep Editing (Ours) |
|---|---|---|---|
| **Focus** | Evaluates unstructured knowledge editing, where knowledge is represented in complex, free-form text rather than structured formats. | Edits event-driven knowledge affecting multiple facts and future tendencies; mirrors how knowledge evolves through events rather than isolated fact changes. | Evaluates deep editing to prevent fact deducibility and preserve broader context; assesses editing techniques via multi-step sequential inference. |
| **Primary Benchmark** | UnKEBench | ELKEN | KnowGIC |
| **Dataset Creation** | 1,000 counterfactual unstructured texts sourced from ConflictQA. | Built on Wikidata for 1,515 event edits; GPT-3.5 paraphrasing and human verification for 6,449 factual and 10,150 tendency questions. | 1,406 reasoning chains; each sample includes 1-5 multi-step questions. |
| **Metrics Used** | FactScore, MMLU, Rouge Scores, BERT Scores, BLEU | Reliability and Locality | IFR, Preservation |

**Efficacy**. This metric quantifies the proportion of cases where the edited output $o_i$ is more likely than the original output $o_c^i$ when the model is queried with the prompt $(s_i, r_i)$:

$$E_i(P_{f_\theta}[o_i|(s_i, r_i)] > P_{f_\theta}[o_c^i|(s_i, r_i)]). \tag{8}$$

**Generalization**. Generalization measures the proportion of cases where $o_i$ remains more probable than $o_c^i$ in paraphrased prompts $N((s_i, r_i))$, which rephrase the original statement while preserving its meaning:

$$E_i(P_{f_\theta}[o_i|N((s_i, r_i))] > P_{f_\theta}[o_c^i|N((s_i, r_i))]) \tag{9}$$

**Specificity**. Specificity evaluates the model's ability to preserve correct facts in neighborhood prompts $O((s_i, r_i))$, which are prompts about related but distinct subjects:

$$E_i(P_{f_\theta}[o_i|O((s_i, r_i))] > P_{f_\theta}[o_c^i|O((s_i, r_i))]) \tag{10}$$

**Fluency**. Fluency assesses the quality of the model's generated text by measuring excessive repetition through the entropy of n-gram distributions. It is computed as:

$$-\frac{2}{3}\sum_k g_2(k)\log_2 g_2(k) + \frac{4}{3}\sum_k g_3(k)\log_2 g_3(k) \tag{11}$$

where $g_n(\cdot)$ represents the frequency distribution of n-grams in the output.

**Consistency**. Consistency evaluates the alignment between the model's generated text and a ground-truth reference. Given a subject $s$, the model $f_\theta$ generates text, and the cosine similarity is computed between the TF-IDF vectors of this output and a reference Wikipedia text about the target $o$, ensuring factual coherence.

## E  Various Prompts used in ThinkEval

The THINKEVAL framework employs a structured set of prompts to extract, evaluate, and edit the internal knowledge of LLMs, as illustrated in Figures 11, 12 and 13. The Fact Extraction Prompt systematically

---

**Triplet-based Question Generation Prompt**

Given a relation triplet which contains the entity, relationship and attribute, generate a question about the entity that would help elicit its relationship with the attribute. Strictly generate the output only.
Example:
Harry Potter, studied at, Hogwarts School of Witchcraft and Wizardry
Output:
Where did Harry Potter study?

Triplet: <target triplet>

---

Figure 11: The template of the triplet-based query generation prompt.

---

**Chain-based Question Generation Prompt**

Given a chain of alternating entities and relationships, generate a question based on all the entities and their relationships so that the answer is the last entity in the chain. Strictly generate the output only.
Example:
Harry Potter, subject, Transfiguration, taught by, Minerva McGonagall
Question:
Who teaches Transfiguration to Harry Potter?

Chain: <target chain>

---

Figure 12: The template of the chain-based query generation prompt.

---

**Fact Extraction Prompt**

You are a fact extraction assistant specialized in parsing Q&A data. Your task is to analyze the input and extract all fact triples (subject, relationship, object) in a precise manner from the given Question and Answer. Each triple should encapsulate a single atomic fact expressed in the text. Focus on characters, events, places. Give a list but not a numbered list.
Example:
Benedetto Varchi, profession, poet
Benedetto Varchi, work, Storia Fiorentina

Question: < question prompt>
Response: <model response>

---

Figure 13: The template of the fact extraction prompt.

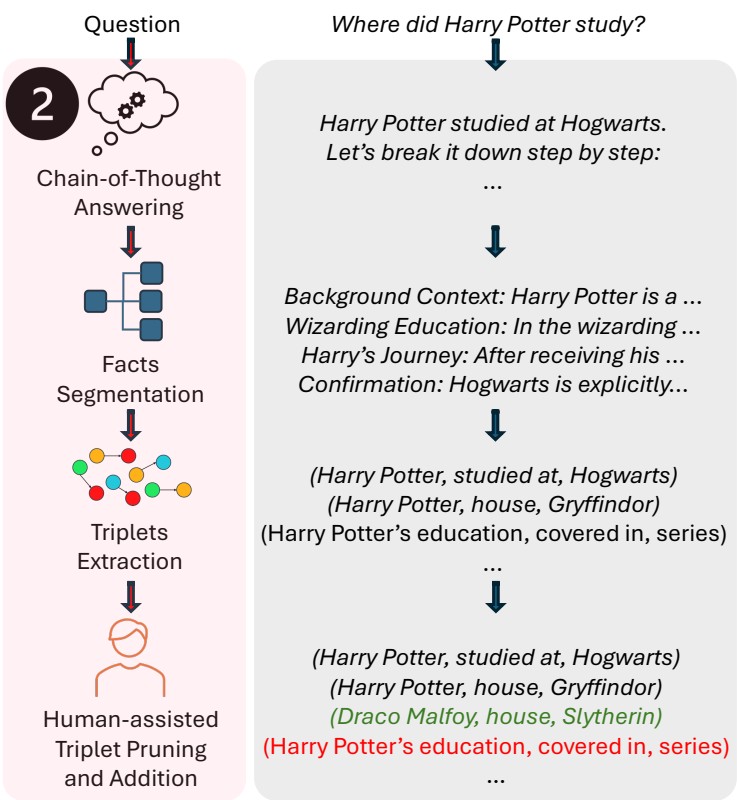

Figure 14: Automated Triplet Generation.

retrieves factual relationships from the model, enabling the construction of its internal knowledge graph by identifying entities and their connections (e.g., mapping relationships like `"Harry Potter, school, Hogwarts"`). The Triplet-Based Query Generation Prompt creates targeted queries from (subject, relation, object) triplets to evaluate factual consistency, such as `"Who teaches Ancient Runes to Hermione Granger?"` for `(Hermione Granger, taught Ancient Runes by, Bathsheda Babbling)`. Similarly, the Chain-Based Query Generation Prompt does the same for a relationship chain. By leveraging these prompts, THINKEVAL effectively maps model-specific knowledge structures, identifies inconsistencies, and facilitates precise evaluation of model editing, thereby addressing deficiencies in LLMs with greater accuracy and reliability.

## F   Explanation of Automated Triplet Generation Process

Figure 14 illustrates the Automated Triplet Generation process, which constructs knowledge graphs by transforming a query like `"Where did Harry Potter study?"` into structured triplets. It begins with a query, followed by CoT reasoning to systematically answer it, such as covering background context, wizarding education, `Harry's` journey, and confirmation that `Hogwarts` is his school. From this, facts are extracted (e.g., `"Harry Potter studied at Hogwarts"`) and converted into triplets: `(Harry Potter, studied at, Hogwarts)`, `(Harry Potter, house, Gryffindor)`, and `(Harry Potter's education, covered in, series)`. Human-assisted pruning removes inaccuracies and irrelevant facts like `(Harry Potter's education, covered in, series)`, retaining the relevant set: `(Harry Potter, studied at, Hogwarts)`, `(Harry Potter, house, Gryffindor)`. Human-assisted triplet addition is utilised to add triplets which may be missed by the automated process even after a few iterations (such as `(Draco Malfoy, house, Slytherin)`). This process, combining CoT reasoning and human oversight, ensures accurate triplets for evaluating fact deducibility and contextual knowledge preservation.

## G    KnowGIC Benchmark Details

Table 6: Various base query templates, triplets, and categories in the KnowGIC benchmark.

| Category | Base query templates | Sample triplet |
|---|---|---|
| Country of Citizenship | _____ is a citizen of | (Byron Dorgan, country of citizenship, USA) |
| Parental Relationship | _____'s child is | (Elvis Presley, child, Lisa Marie Presley) |
| Creation and Origin | _____ was created by | (Miss Piggy, created by, Jim Henson) |
| Marital Relationship | _____ is married to | (Victoria Beckham, spouse, David Beckham) |
| Sport Affiliation | _____ is associated with the sport of | (Mark Teixeira, sport, baseball) |
| Country of Origin | _____ was created in the country of | (basketball, country of origin, USA) |
| Development Origin | _____ was developed by | (Atari Jaguar, developer, Atari Corporation) |
| Work Location | _____ worked in the city of | (Vincent Auriol, work location, Paris) |
| Employers | _____ was employed by | (Ward Kimball, employer, The Walt Disney Company) |
| Production Entity | The company that produced _____ is | (Buick LaCrosse, produced by, General Motors) |
| Study Location | _____ studied at | (Harry Potter, school, Hogwarts) |

To provide an overview of the KnowGIC benchmark employed in our evaluation of model-editing techniques, Table 6 presents the base queries, their corresponding triplet samples, and their respective categories. Our evaluation includes a total of 83 distinct relationship types, derived from extending these base relationships through controlled logical transformations. The dataset encompasses 1,406 samples, each being an implication chain of varying lengths (1 to 5 steps), as detailed in Table 1. Table 7 provides a few samples of the implication chains of varying lengths present in the dataset. It spans diverse knowledge domains, including country of citizenship (e.g., "What is the country of citizenship of Byron Dorgan?"), parental and marital relationships (e.g., "Who is Hillary Clinton's child?" and "Who is Nicole Richie married to?"), creation origins (e.g., "Who was Miss Piggy created by?"), sport affiliations (e.g., "Which sport is Mark Teixeira associated with?"), country of origin (e.g., "Which country was Bakersfield sound created in?"), work locations (e.g., "Which city did Petronius work in?"), production entities (e.g., "Which company is Buick LaCrosse produced by?"), and study locations (e.g., "Where did Harry Potter study?"), among others.

Each triplet represents a factual relationship that serves as the foundation for both direct queries and multi-step implication chains, which are designed to test fact deducibility and broader contextual knowledge preservation post-edit. For example, (`Harry Potter, school, Hogwarts`) is evaluated through the direct query `"Where did Harry Potter study?"` and extended into implication chains such as those exploring related entities like schoolmates and teachers. This structured representation enables a systematic evaluation of knowledge consistency across diverse domains, facilitating the evaluation of editing techniques' effectiveness in reducing deducibility (via IFR) and preserving connected knowledge (via Preservation), thus supporting the development and refinement of frameworks like THINKEVAL for robust editing in LLMs. Figures 15, 16, 17, 18, 19, 20, 21 and 22 present a few of the knowledge graphs present in KnowGIC, which can be further extended using THINKEVAL.

## H    Harry Potter Case-Study Details

Table 8 provides an overview of comparison of the existing metrics with our proposed metrics IFR, focussing on capability to evaluate indirect fact extraction. We conduct a case-study on the sample (`Harry Potter, school, Hogwarts`) from KnowGIC to evaluate the need for new metrics in deep editing, aiming to edit the fact (Hogwarts → Ilvermorny). We evaluated the edit using Efficacy and our proposed IFR metric. This

Table 7: Multi-step implication chains samples from KnowGIC. Each row presents an implication chain with the request integrated into the sample queries to evaluate fact deducibility and broader contextual integrity post-edit.

| $n$-step | Implication chain with knowledge triplet |
|---|---|
| 1 | **Triplet**: *(Petronius, work location, Rome)* 
 In which city did Petronius work? |
| 2 | **Triplet**: *(Hillary Clinton, child, Chelsea Clinton)* 
 Who is Hillary Clinton's son-in-law? → Who is Marc Mezvinsky's wife? |
| 3 | **Triplet**: *(Ward Kimball, employer, Walt Disney)* 
 Which character did Ward Kimball animate? → In which film does Jiminy Cricket appear? → Which company produced Pinocchio? |
| 4 | **Triplet**: *(Harry Potter, school, Hogwarts)* 
 Who were Harry Potter's schoolmates? → Who were Ron Weasley's schoolmates? → Who taught Transfiguration to Hermione Granger? → Which school is Professor McGonagall the headmistress of? |
| 5 | **Triplet**: *(Bakersfield sound, country of origin, USA)* 
 Which city is known for the creation of the Bakersfield sound? → In which valley is Bakersfield situated? → In which county is the San Joaquin Valley located? → In which state is Kern County situated? → What country is California located in? |

Table 8: Comparison of model editing metrics. This table compares existing metrics with our proposed IFR, highlighting its focus on addressing indirect fact extraction.

| Metric | What It Measures | Quantifies Indirect Fact Extraction |
|---|---|---|
| Efficacy | Success in outputting the edited fact for direct queries. | ✗ |
| Generalization | Performance on rephrased versions of the edited query. | ✗ |
| Specificity | Ensures unrelated queries are unaffected by the edit. | ✗ |
| Fluency | Naturalness and coherence of the model's responses post-edit. | ✗ |
| Consistency | Logical coherence across related facts. | ✓ (Partially—focuses on direct consistency) |
| IFR | Extent to which the original fact remains deducible through multi-step reasoning. | ✓ |
| Preservation | Retention of facts related to the edited subject but not directly targeted. | ✗ |

case-study highlights that traditional metrics like Efficacy are insufficient for deep editing tasks. IFR's focus on multi-step chains provides a more robust evaluation. These findings stress the need for advanced metrics and better editing methods for comprehensive knowledge updates in language models.

## I  Generalization beyond factual model editing

While our work focuses on factual model edits, which provide a well-defined and controllable setting for evaluating indirect knowledge leakage, the core design of THINKEVAL is not inherently limited to factual knowledge alone. THINKEVAL relies on CoT-reasoning and graph-based causal modeling, both of which can, in principle, be extended to other forms of knowledge.

For example, in commonsense or procedural reasoning scenarios, graph nodes may represent abstract concepts, actions, or intermediate states rather than factual entities. Edges extracted from CoT traces can encode causal, temporal, or conditional ("if–then") relationships, enabling the construction of thought-based

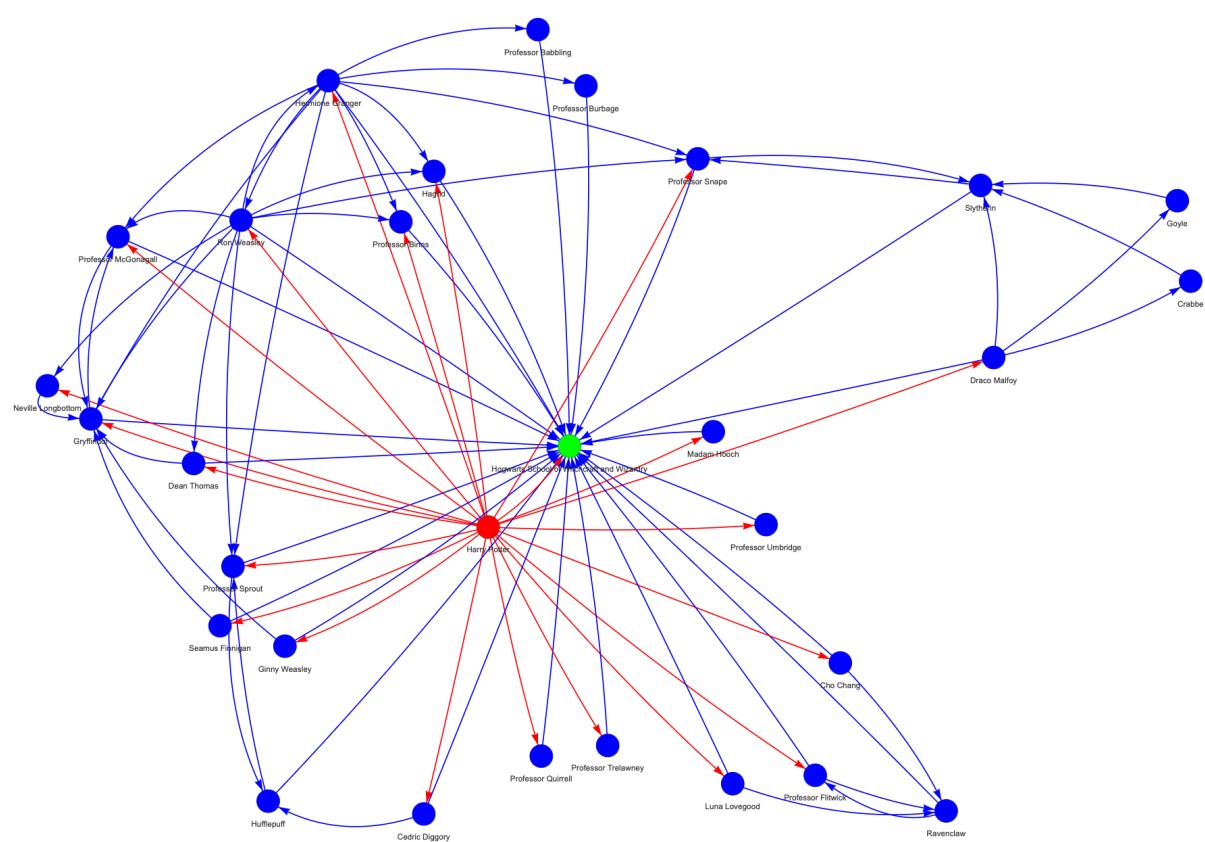

Figure 15: Sample of a constructed knowledge graph from the KnowGIC implication chains for the relationship triplet (Harry Potter, school, Hogwarts). The graph was expanded with the generated triplets until 100 implications paths from the initial subject to object were achieved. The red node represents the initial triplet subject, and the green node represents the initial triplet object.

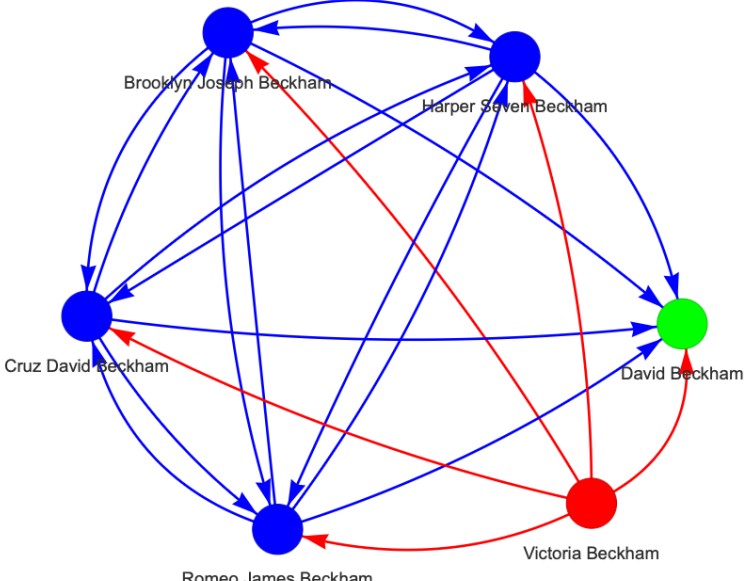

Figure 16: Sample of a constructed knowledge graph from the KnowGIC implication chains for the relationship triplet (Victoria Beckham, spouse, David Beckham).

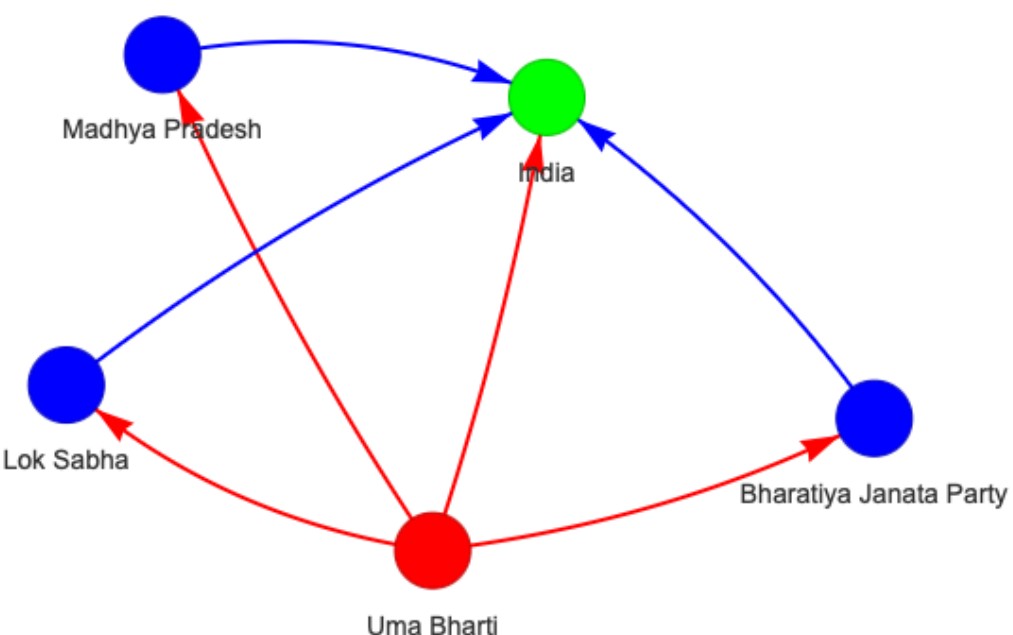

Figure 17: Sample of a constructed knowledge graph from the KnowGIC implication chains for the relationship triplet (Uma Bharti, country of citizenship, India).

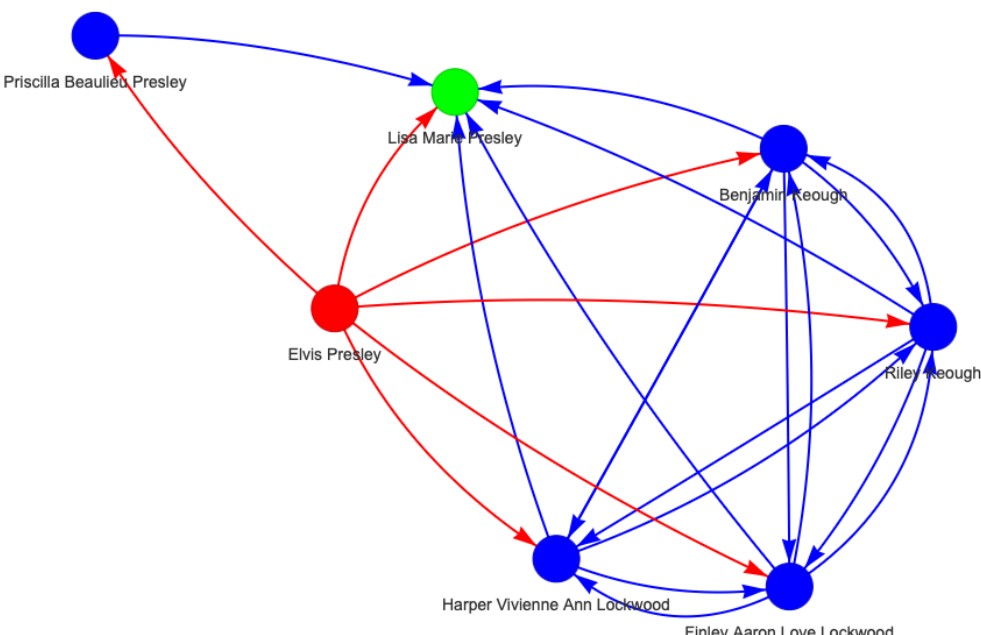

Figure 18: Sample of a constructed knowledge graph from the KnowGIC implication chains for the relationship triplet (Elvis Presley, child, Lisa Marie Presley).

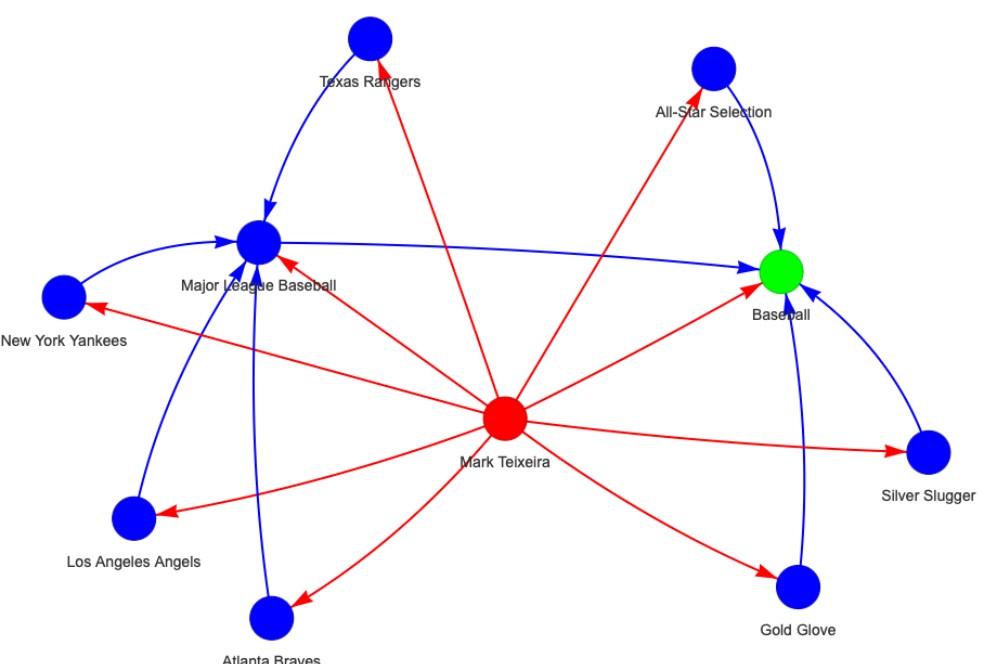

Figure 19: Sample of a constructed knowledge graph from the KnowGIC implication chains for the relationship triplet (Mark Teixeira, sport, baseball).

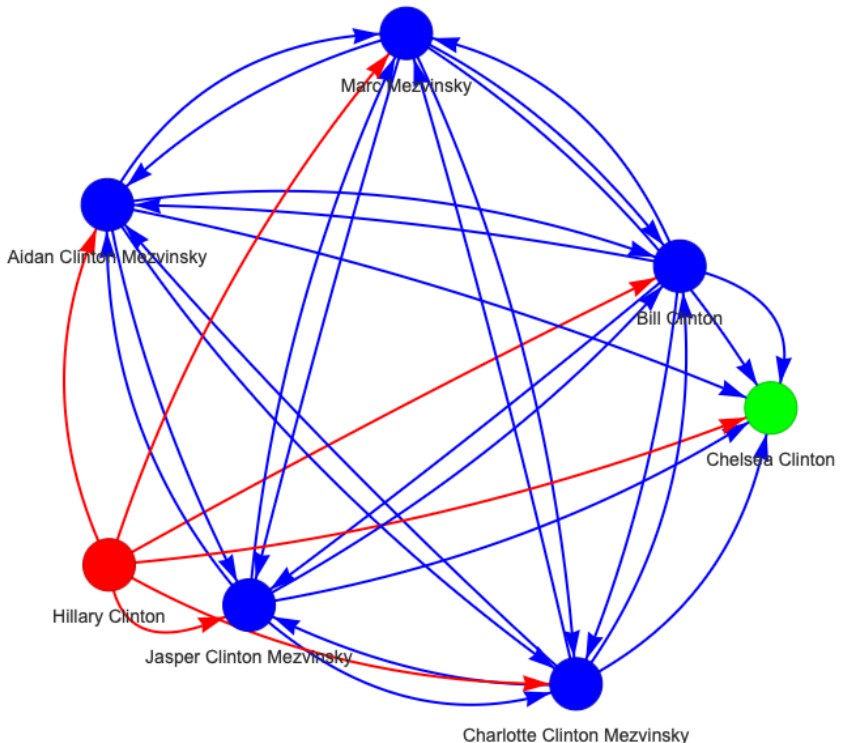

Figure 20: Sample of a constructed knowledge graph from the KnowGIC implication chains for the relationship triplet (Hillary Clinton, child, Chelsea Clinton).

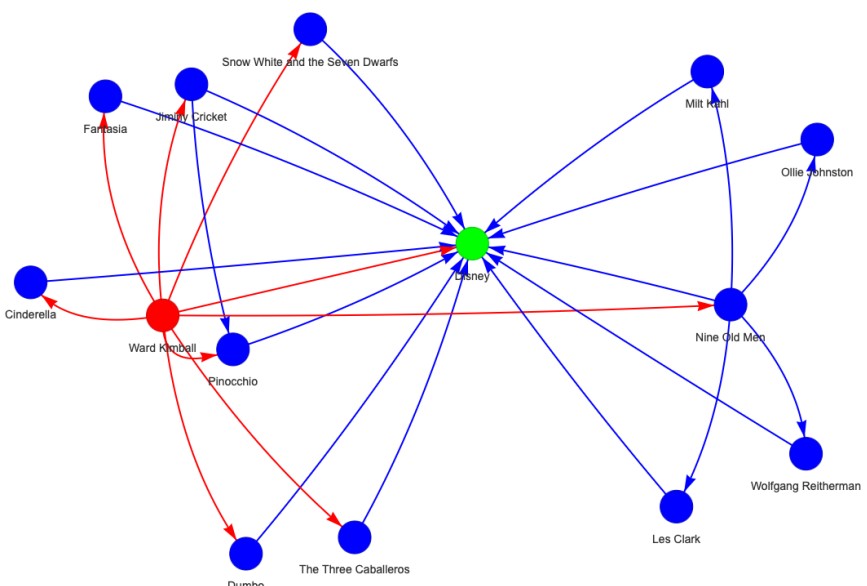

Figure 21: Sample of a constructed knowledge graph from the KnowGIC implication chains for the relationship triplet (Ward Kimball, employer, The Walt Disney Company).

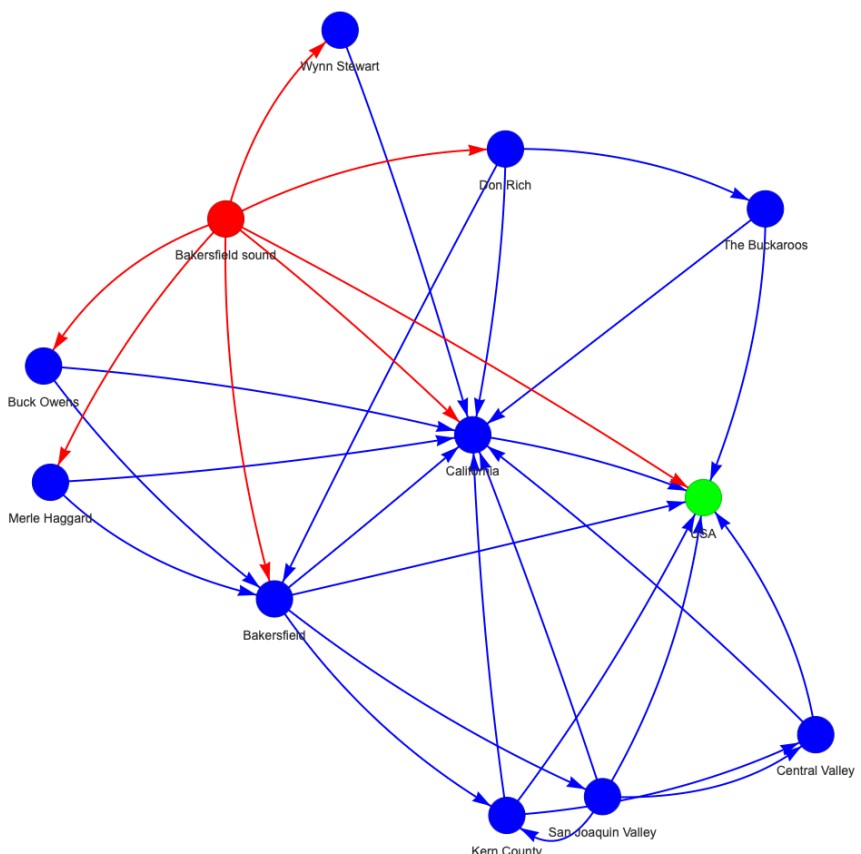

Figure 22: Sample of a constructed knowledge graph from the KnowGIC implication chains for the relationship triplet (Bakersfield sound, country of origin, USA).

graphs over non-factual knowledge structures. Similarly, for rule-based or logical knowledge, nodes may correspond to predicates or rules, with edges capturing inference dependencies.

Under such extensions, the same evaluation paradigm (measuring the persistence or disruption of reasoning paths after targeted edits) can be used to assess indirect leakage and preservation behavior beyond factual settings. We leave the systematic instantiation and benchmarking of THINKEVAL for these broader knowledge types as an important direction for future work.

## J Illustrative example of IFR calculation

To provide clarity on the IFR metric, we present a step-by-step calculation using a representative 3-step reasoning chain. This example illustrates how the metric quantifies the persistence of an original fact through indirect reasoning paths after a model edit has been performed.

### J.1 Scenario Setup

Consider a scenario where we attempt to edit a specific fact in the model's knowledge base:

- **Target Edit:** Harry Potter studied at "Hogwarts" → "Ilvermorny".

Despite this edit, the original fact (*Harry Potter → Hogwarts*) may still be recoverable if the model retains related knowledge that allows it to reconstruct the original connection through multi-step reasoning.

### J.2 Sample Reasoning Chain

We define a chain of $n = 3$ questions that logically lead back to the original fact:

1. $Q_1$: Who were Harry Potter's schoolmates? → $A_1$: Ron Weasley.

2. $Q_2$: Which house is Ron Weasley in? → $A_2$: Gryffindor.

3. $Q_3$: Which school does Gryffindor belong to? → $A_3$: Hogwarts.

### J.3 Chain Recovery and IFR Computation

We measure the model's confidence (probability) in producing the correct intermediate facts both before and after the edit:

| Question | Pre-Edit Probability ($P_{pre}$) | Post-Edit Probability ($P_{post}$) |
|---|---|---|
| $C_1Q_1$ | 0.90 | 0.70 |
| $C_1Q_2$ | 0.85 | 0.80 |
| $C_1Q_3$ | 0.90 | 0.85 |

Table 9: Pre-edit and post-edit probabilities for a 3-step reasoning chain.

**Step 1: Calculate Chain Recovery ($R$)**
The chain recovery represents the overall likelihood that the original fact can be reconstructed through the reasoning path, calculated as the product of the link probabilities:

$$R_{pre} = 0.9 \times 0.85 \times 0.9 = 0.69$$

$$R_{post} = 0.7 \times 0.8 \times 0.85 = 0.48$$

.

**Step 2: Apply the IFR Formula**
The IFR metric normalizes the change in recovery relative to the chain length $n$:

$$\text{IFR} = \frac{(R_{\text{post}}/R_{\text{pre}})/\sqrt{n}}{1/\sqrt{n}} = \frac{0.48/\sqrt{3}}{0.69/\sqrt{3}} \approx 0.70$$

### J.4 Interpretation

In this instance, an IFR of 0.70 indicates that even after the model was explicitly edited to believe Harry Potter attended Ilvermorny, the original association with Hogwarts remains 70% recoverable with respect to the original model. This demonstrates the knowledge leakage that THINKEVAL is designed to systematically detect. The $(1/\sqrt{n})$ weighting normalizes for chain length by mitigating the compounding uncertainty of multi-step reasoning. This prevents long, noisy chains from dominating the metric and emphasizes shorter, higher-confidence reasoning paths.

Furthermore, both IFR and Preservation are inherently normalized with respect to each model's pre-edit factual accuracy. Specifically, the pre-edit performance terms in the denominators of each serve as a calibration factor, ensuring that post-edit scores are interpreted relative to each model's original knowledge baseline. This design mitigates model-specific bias and enables fair comparison across models with differing pre-edit accuracies.

## K  Scalability and automated validation study

To evaluate the practical deployment of THINKEVAL, we conducted a scalability study measuring the end-to-end runtime and the efficacy of an automated LLM-based judge to reduce human overhead. We have provided all the samples in our repository.

### K.1  Computational Efficiency

We measured the average end-to-end runtime of the THINKEVAL process over incremental iterations until a threshold of at least 100 samples was reached. The initial setup began with a single triplet. The growth of the reasoning graph and associated costs are detailed in Table 10.

| Iteration | Triplets | Paths (Subject $\rightarrow$ Target) | CoT Queries |
|:---:|:---:|:---:|:---:|
| 1 | 5 | 1 | 12 |
| 2 | 15 | 3 | 43 |
| 3 | 23 | 10 | 124 |
| 4 | 51 | 27 | 405 |
| 5 | 83 | 133 | 654 |

Table 10: Growth of reasoning paths and computational queries across iterations.

The empirical results confirm the efficiency of the framework:

- **Average CoT Runtime:** 4.15 seconds per prompt.

- **Total Computational Cost:** ~8.4 minutes of CoT GPU time for 133 samples.

- **End-to-End Latency:** ~15 minutes total, including human verification (~4–5 min) and graph synthesis.

These findings suggest that while CoT-prompting is the primary computational driver, the validation and graph synthesis stages remain lightweight and highly parallelizable.

### K.2  LLM-based Automated Judging

To minimize human dependency, we implemented an LLM-based judge (using `GPT-4o`) to validate and prune triplets. The judge filters misinformation to ensure only high-quality triplets are used for graph construction. We evaluated the judge's performance across five iterations, as summarized in Table 11.

The LLM-judge achieved a precision of 88%, a recall of 86%, and an F1-score of 87% (based on aggregate totals of $TP = 71, FP = 10, FN = 12$). Notably, the automated judge reached the 100-reasoning-chain

| Iteration | True Positives (TP) | False Positives (FP) | False Negatives (FN) | Correct Paths |
|---|---|---|---|---|
| 1 | 5 | 0 | 0 | 1 |
| 2 | 13 | 1 | 2 | 3 |
| 3 | 20 | 4 | 3 | 8 |
| 4 | 48 | 7 | 2 | 35 |
| 5 | 71 | 10 | 12 | 109 |

Table 11: Performance of the `GPT-4o` judge in identifying valid triplets and reasoning paths.

threshold in the same number of iterations (five) as the human-verified baseline. Our results indicate that reserving human effort for final verification while using the LLM-judge for intermediate pruning can reduce human validation effort by 70–75% with only marginal loss in accuracy.

## L   Evaluation of parameter-preserving model editing

To assess the robustness of indirect knowledge preservation in non-parametric editing scenarios, we evaluated the In-Context Knowledge Editing (IKE) technique (Zheng et al. (2023)) on the KnowGIC dataset. Unlike traditional parameter-modifying techniques, IKE relies on in-context learning to update model behavior without altering underlying parameters. The results of this evaluation are summarized in Table 12.

| Model | Preservation | IFR |
|---|---|---|
| GPT2-XL | 0.680 | 0.542 |
| Llama3-8B | 0.729 | 0.756 |
| Qwen2.5-7B | 0.623 | 0.247 |

Table 12: Evaluation of IKE across different LLMs using the KnowGIC dataset.

The empirical findings indicate that IFR remains a significant factor even within parameter-preserving techniques. For instance, Llama3-8B exhibits an IFR of 0.756, suggesting that a substantial portion of the original knowledge remains accessible through reasoning paths despite the presence of an in-context edit. These results suggest that in-context editing techniques do not fully prevent hidden factual leakage, highlighting the need for more rigorous editing strategies.

## M   Responsible model editing and future directions

Responsible model editing in LLMs necessitates a rigorous ethical framework to balance utility with safety. Techniques like THINKEVAL primarily serve to enhance model reliability by facilitating privacy protection, such as implementing the "right to be forgotten" for sensitive data, and the rapid correction of misinformation. However, model editing could be leveraged for malicious injection of bias or the suppression of legitimate information. To mitigate these risks, we advocate for the implementation of transparent edit logging and strict human-in-the-loop verification for all high-stakes or sensitive modifications. Furthermore, as our results demonstrate that single-query evaluations often fail to reveal hidden inference pathways, we propose that robust sequential testing be a mandatory component of the safety audit process for any edited model to prevent the unintended leakage of suppressed facts.

Achieving low IFR while maintaining high Preservation remains a central challenge for model editing. Future editing techniques may benefit from explicitly optimizing this trade-off by incorporating constraints or regularizers that minimize indirect leakage while bounding degradation on unrelated knowledge. In addition, integrating multi-step and adversarial reasoning probes during the editing process can help proactively close indirect inference pathways that single-query checks often miss. In this sense, THINKEVAL may serve not only as an evaluation framework, but also as a practical guide for developing editing techniques that balance factual precision with contextual retention, supporting more robust and responsible model editing.

## N    Reproducibility and transparency

To support reproducibility and transparency, we release all the model versions and editing configurations utilised in our experiments. All experiments are performed on an NVIDIA A100 80GB GPU.

**Model Versions.** All language models used in our experiments are publicly available on HuggingFace. Unless otherwise stated, models are used in their official, unmodified form as released by their respective authors, and no additional fine-tuning is performed prior to editing.

In our experiments, we experiment over the following models from HuggingFace:

- meta-llama/Meta-Llama-3-8B-Instruct,

- Qwen/Qwen2.5-7B-Instruct,

- openai-community/gpt2-xl.

**Editing Configurations.** For each editing technique, we utilise the implementation given in AlphaEdit's repository (Jiang (2024)). We follow the hyperparameter configurations provided in the publicly released implementation, including the choice of layers to edit, thus ensuring faithful comparison.

## O    Dataset reliability

To ensure the robustness of our evaluation framework and address potential biases in human validation, we conducted a pilot annotation study with three research group members. A random sample of 250 implication chains was reviewed to verify the presence of logical implications. While the strength of implication may decrease as chains grow longer, we mitigate this by scaling results by $1/\sqrt{n}$, where $n$ represents the number of links in the chain; this weighting strategy was unanimously agreed upon by the annotators.

Furthermore, all factual triples were reviewed. To quantify consistency, we report raw agreement and Gwet's AC1, as the latter provides a more stable reliability metric in the presence of high agreement.

Table 13: Inter-annotator agreement statistics for human validation.

| Metric | Value |
|---|---|
| Raw Agreement | ∼97% |
| Gwet's AC1 | ∼98% |

As shown in Table 13, the results demonstrate high inter-annotator consistency and minimal bias.

## P    Automation and human-in-the-loop integration

To clarify the operational mechanics of THINKEVAL, we define the role of LLM and human involvement within our algorithm. THINKEVAL utilizes a single, unedited model; no secondary LLM agents or external models are employed for meta-processing. The workflow is categorized into LLM-driven automation, rule-based logic, and human-in-the-loop (HITL) refinement, as detailed in Table 14.

## Q    Comparison: ThinkEval vs. MQuAKE

While MQuAKE focuses on multi-hop reasoning within a static dataset, THINKEVAL provides a dynamic framework to evaluate fact leakage and preservation through sequential queries. The fundamental differences are summarized in Table 15.

Table 14: THINKEVAL Automation and HITL Components

| Automation Type | Component | Where in Figure 2 | Comment |
|---|---|---|---|
| Using unedited original LLM | Query generation | Right before blue region ① | Similar to LLM-driven query generation in Zhong et al. (2023) |
| | LLM response generation | Blue region ① | - |
| | Chain-of-Thought answering | Pink region ② | - |
| | Triplets extraction | Pink region ② | Similar to Wang et al. (2025a) and other benchmarks |
| No LLM or human involvement | Knowledge validation | Blue region ① | Rule-based, same way as Zhong et al. (2023) (using alias list) |
| | Fact segmentation | Pink region ② | Segmented into individual statements |
| | Chain sequencing | Green region ③ | - |
| Human-in-the-loop (HITL) | Query refinement | Blue region ① | - |
| | Triplet pruning and addition | Pink region ② | - |
| | Final sanity checks | Dataset (Green region ③) | - |

Table 15: Comparison between MQuAKE and THINKEVAL.

| Dimension | MQuAKE | ThinkEval |
|---|---|---|
| Fundamental difference | Dataset | Framework to create datasets |
| What they evaluate | Multi-hop reasoning in edited LLMs; however, multi-hop reasoning remains beyond current parameter-editing techniques, which still struggle with ripple effects (Wang et al. (2025b)) | Shows edited-out fact leakage via sequential queries; bridges the gap between reasoning and ripple effects by mapping paths in knowledge graphs, thereby highlighting which additional facts require editing and which must be preserved |
| How they evaluate | Multi-hop reasoning questions where multiple relations are compressed into a single query | Multi-step sequential queries; evaluates reasoning across sequences of queries, extending beyond single-query formats to reflect how users may naturally interact with LLMs |
| Adaptability | Unedited models correctly answer only 30–40% of MQuAKE's multi-hop queries (as reported in Zhong et al. (2023)), restricting the usable portion of the dataset | Mitigates this issue by decomposing reasoning into simpler, sequential steps |
| Metrics | Single-query metrics | IFR (Section 5.1) and Preservation (Cohen et al. (2024)) |

## R   Automated workflow robustness and results stability

We conduct a controlled stability experiment to test whether THINKEVAL's key findings remain robust when a secondary LLM is introduced. In particular, we examine whether (i) editing technique rankings and (ii) the characteristic rise-then-fall behavior of IFR persist under automated paraphrasing.

**Experimental Setup.**   We select a 100-chain subset from KnowGIC and introduce a secondary model into the pipeline for automated paraphrase generation. Specifically, we use `GPT-4o` to generate three paraphrases

Table 16: Stability test results across models and techniques using paraphrased query chains.

| Model | Technique | IFR | Pres. | Overall | 1-step | 2-step | 3-step | 4-step |
|---|---|---|---|---|---|---|---|---|
| GPT2-XL | AlphaEdit | 0.736 | 0.903 | 0.736 | 0.433 | 0.773 | 0.824 | 0.652 |
| | MEMIT | 0.727 | 0.880 | 0.727 | 0.433 | 0.903 | 0.714 | 0.671 |
| | PRUNE | 0.250 | 0.782 | 0.250 | 0.311 | 0.280 | 0.248 | 0.232 |
| | RECT | 0.754 | 0.850 | 0.754 | 0.667 | 0.752 | 0.829 | 0.686 |
| | ROME | 0.279 | 0.752 | 0.279 | 0.233 | 0.250 | 0.282 | 0.301 |
| Llama-3-8B | AlphaEdit | 0.678 | 0.900 | 0.678 | 0.342 | 0.604 | 0.683 | 0.781 |
| | MEMIT | 0.800 | 0.911 | 0.800 | 0.615 | 0.867 | 0.814 | 0.767 |
| | PRUNE | 0.309 | 0.647 | 0.309 | 0.111 | 0.434 | 0.363 | 0.170 |
| | RECT | 0.894 | 0.880 | 0.894 | 0.615 | 0.897 | 0.925 | 0.875 |
| | ROME | 0.217 | 0.571 | 0.217 | 0.081 | 0.350 | 0.167 | 0.187 |
| Qwen2.5-7B | AlphaEdit | 0.391 | 0.917 | 0.391 | 0.285 | 0.315 | 0.512 | 0.313 |
| | MEMIT | 0.394 | 0.865 | 0.394 | 0.297 | 0.392 | 0.458 | 0.323 |
| | PRUNE | 0.411 | 0.587 | 0.411 | 0.316 | 0.320 | 0.442 | 0.445 |
| | RECT | 0.525 | 0.823 | 0.525 | 0.387 | 0.492 | 0.558 | 0.520 |
| | ROME | 0.261 | 0.537 | 0.261 | 0.198 | 0.237 | 0.312 | 0.221 |

for each atomic factual triple (e.g., *(Harry Potter, school, Hogwarts)* → "Harry Potter studied at", "Harry Potter's school is", "Harry Potter went to school at").

All five editing techniques are evaluated across three base models: GPT-2-XL, Llama-3-8B, and Qwen2.5-7B. To incorporate paraphrasing into the evaluation, each link in a reasoning chain is scored by averaging model correctness probabilities across its three paraphrased queries. IFR and Preservation are then recomputed using these averaged scores, ensuring that each chain reflects robustness across surface-level linguistic variation. For transparency and reproducibility, we release the full paraphrased subset along with resultant plots in our repository.

Table 16 summarizes IFR and Preservation scores under paraphrasing across all models and editing techniques. We observe strong stability across all dimensions:

- **Preservation of relative rankings.** Across all three models, the relative ordering of editing techniques remains consistent with the original evaluation. AlphaEdit, MEMIT, and RECT continue to exhibit higher IFR alongside strong Preservation, while ROME and PRUNE consistently achieve lower IFR at the cost of reduced Preservation. This mirrors the same trade-off pattern reported in the main experiments.

- **Persistence of IFR dynamics.** The characteristic rise-then-fall behavior of IFR as chain length increases persists under paraphrasing, indicating that this effect is not an artifact of specific prompt formulations but reflects stable reasoning behavior in edited models.

- **Robustness to automated workflow choices.** Introducing a secondary LLM for paraphrase generation does not alter the qualitative conclusions drawn from THINKEVAL. IFR, Preservation, and editing-technique comparisons remain stable, suggesting that THINKEVAL is robust to reasonable automation choices and workflow variations.

These results demonstrate that THINKEVAL is not sensitive to surface-level phrasing or evaluation pipeline details, reinforcing its reliability as an evaluation framework for model editing.

The design of THINKEVAL is deeply informed by the broader literature on maintaining representational consistency and mitigating catastrophic forgetting. Specifically, our focus on minimizing IFR while maximizing Preservation aligns with the principles of consistency alignment and calibration (Gao et al. (2025a;b)). These works emphasize that as LLMs are specialized or edited, maintaining the underlying structural coherence of their embedding space is critical to preventing the degradation of previously learned knowledge. THINKEVAL extends this concept by providing an evaluation framework that explicitly tests whether this

coherence survives multi-step, sequential query chains. Furthermore, the practical utility of our framework extends into high-stakes, specialized domains. As demonstrated in applications like medical scribing for specialized clinical notes (Goyal et al. (2025)), the ability to edit models while ensuring they do not leak suppressed or outdated medical facts is paramount. By integrating THINKEVAL into the lifecycle of domain-specific model adaptation, practitioners can better assess the robustness of their edits in environments where factual precision and logical consistency are non-negotiable.

