# OpenReview forum: "ThinkEval: Practical Evaluation of Knowledge Leakage in LLM Editing using Thought-based Knowledge Graphs"
_TMLR — Accepted by TMLR_

### Review · Reviewer_jMMQ · 2025-10-09

**Summary Of Contributions:**

Paper Summary

This paper introduces ThinkEval, a framework for systematically evaluating how well model-editing techniques update and preserve knowledge in large language models.
The authors identify an important problem indirect fact leakage, where “edited-out” knowledge can still be inferred through multi-step reasoning.
ThinkEval's core contributions are:
1. Deep Editing: A new evaluation standard that requires an edit to be successful not just for direct queries, but also against inference through chains of related facts.
2. Indirect Fact Recovery (IFR): A new metric that quantifies how much of the original fact can be recovered via multi-step reasoning after an edit.
3. The KnowGIC Benchmark: A dataset of 1406 multi-step reasoning paths built to test editing techniques under the deep editing setting.
The study provides detailed experiments across five state-of-the-art editing methods (ROME, MEMIT, PRUNE, RECT, AlphaEdit) and three LLMs (Llama3-8B, Qwen2.5-7B, GPT2-XL), analyzing the trade-offs between IFR, preservation, and efficacy.

Summary Of Strengths
1. Novel and Important Problem Formulation: The paper successfully identifies and formalizes a significant, overlooked issue in model editing: indirect fact recovery via multi-step reasoning. The "Harry Potter" case study is a compelling and clear illustration of the problem.
2. Comprehensive Framework (ThinkEval): ThinkEval is a well-designed, multi-component framework that integrates CoT reasoning, automated triplet generation, human-in-the-loop validation, and graph synthesis. Its cyclical, iterative nature is a strength for building robust, model-specific evaluations.
3. Innovative "Deep Editing" Paradigm and IFR Metric: The concepts of "deep editing" and the IFR metric are major contributions. They provide a principled way to measure editing success beyond direct query performance, addressing a clear gap in existing evaluation methodologies.

Summary Of Weaknesses
1. Scalability and Computational Cost: The ThinkEval process involves multiple rounds of CoT prompting, human validation, and graph synthesis, which could become computationally expensive. However, the paper provides no quantitative data on runtime, GPU hours, or human effort.
2. Model-Specific Knowledge Gaps: Appendix A highlights a significant issue: the KnowGIC benchmark, built using Llama3-8B, contains facts that Qwen2.5-7B does not know. This raises questions about the fairness and universality of the benchmark when evaluating different models and suggests that a perfectly model-agnostic benchmark might be infeasible, though the authors rightly frame ThinkEval as a model-specific tool.
3. Limited Scope of Editing Techniques: The evaluation focuses exclusively on parameter-modifying techniques. Including and contrasting with prominent parameter-preserving methods (e.g., SERAC, IKE) would have provided a more complete picture of the editing landscape and how different architectural choices impact indirect leakage.
4. Clarity on IFR Calculation: The IFR formula (Equation 6) is complex. While the intuition is clear, a more detailed explanation or a simplified illustrative example in the main text would improve accessibility for a broader audience. The weighting by 1/√n_i is justified but could be further motivated.
5. Lack of Discussion on Generalization Beyond Factual Edits: The paper primarily focuses on factual edits involving entity-level knowledge. It remains unclear whether ThinkEval can generalize to other types of edits, such as commonsense reasoning, procedural knowledge, or logical rules, which limits the framework’s perceived versatility.
6. Human Evaluation Bias and Consistency: Given that human reviewers are involved in validating and pruning reasoning chains, the paper lacks an analysis of inter-annotator agreement or measures to ensure consistency. Potential human bias or subjectivity could influence the benchmark’s reliability.
7. Absence of Ethical and Responsible Editing Discussion: The paper does not address the ethical implications of model editing, such as the potential misuse of editing techniques to distort factual knowledge or introduce bias. A short discussion on responsible use and mitigation strategies would enhance the paper’s completeness and societal relevance.
8. Reproducibility and Transparency Limitations: The paper provides limited details on experimental reproducibility. Key implementation factors such as model versions, prompt templates, or hyperparameter settings are not described in sufficient depth, which may hinder replication and fair comparison by other researchers.

**Audience:**

Yes

**Audience Explanation:**

The paper introduces ThinkEval, a novel framework for assessing indirect knowledge leakage—a problem that is both timely and underexplored within the LLM community. By defining the deep editing paradigm and proposing the Indirect Fact Recovery metric, the study contributes a new and practical perspective on how to evaluate editing consistency and knowledge preservation beyond direct factual correctness. Additionally, the construction of the KnowGIC benchmark, which incorporates multi-step reasoning chains, extends current evaluation methodologies in a way that bridges causal reasoning and factual consistency—topics that are central to TMLR’s readership.

**Broader Impact Concerns:**

While the paper makes a valuable methodological contribution, there are several broader impact concerns that warrant deeper discussion. Specifically, model editing—the central focus of this work—has the potential for both beneficial and harmful applications. The ability to systematically modify knowledge in large language models could be misused to distort factual information, reinforce biases, or erase historically or socially significant content. Although the paper focuses on evaluating editing accuracy and preservation, it does not sufficiently address the ethical responsibilities and safeguards that should accompany such powerful interventions.

**Claims And Evidence:**

Yes

**Claims Explanation:**

The paper presents a well-structured framework, ThinkEval, which is validated through extensive experiments involving five prominent model-editing techniques (ROME, MEMIT, PRUNE, RECT, and AlphaEdit) and three large language models. The proposed Indirect Fact Recovery metric and Preservation measure are mathematically defined and consistently applied throughout the evaluation. The results are well-illustrated with both quantitative tables and qualitative case studies, which effectively substantiate the authors’ core claim that existing model-editing techniques fail to suppress indirect knowledge leakage.

**Requested Changes:**

1. Automation for Scalability(Critical): Investigate ways to reduce human dependency in the ThinkEval loop, perhaps by using a more powerful "judge" LLM for initial triplet validation and pruning, reserving human effort for final verification. A study on the trade-off between automation and accuracy would be valuable.
2. Benchmark Robustness and Model-Specific Bias(Critical): To address model-specific knowledge discrepancies (as observed between Llama3-8B and Qwen2.5-7B), consider constructing a core benchmark subset verified to be known by multiple base models. Alternatively, introduce a calibration or normalization step that adjusts IFR and Preservation scores based on each model’s pre-edit factual accuracy. This would make comparisons across models more interpretable and fair.
3. Discussion on Mitigation Strategies(Recommended): The paper excellently diagnoses the problem. A discussion or preliminary ideas on how future editing techniques could be designed to achieve low IFR and high Preservation simultaneously would be a valuable addition, positioning ThinkEval not just as an evaluator but as a guide for future development.
4. Broaden Generalization Discussion(Recommended): To increase the framework’s perceived versatility, the authors should discuss whether ThinkEval can be generalized beyond factual edits, such as those involving commonsense reasoning, procedural knowledge, or logical rules. Addressing how the framework can be applied to other knowledge types would demonstrate its broader applicability and strengthen its potential use in various real-world scenarios, including those requiring higher-level cognitive reasoning.
5. Human Consistency and Ethical Considerations(Recommended): The paper would benefit from a more detailed discussion of human reviewer bias and consistency, especially given the role of human validation in the ThinkEval process. The authors should explore inter-annotator agreement and how human subjectivity might influence the evaluation outcomes. Additionally, a brief section addressing the ethical implications of model editing—such as the risk of misuse for manipulating knowledge—would add value. Including responsible usage guidelines or mitigation strategies would align the paper with growing concerns over ethical AI development.
6. (Recommended)To strengthen the theoretical foundation and situate this work more clearly within the literature on editing robustness and consistency preservation, the authors are encouraged to cite related studies such as “Mitigating Forgetting in Adapting Pre-trained Language Models to Text Processing Tasks via Consistency Alignment” and “Shaping Pre-trained Language Models for Task-specific Embedding Generation via Consistency Calibration.” These works address the challenges of catastrophic forgetting and representation inconsistency during model adaptation, which are conceptually aligned with ThinkEval’s focus on maintaining knowledge preservation and reducing IFR after model editing. In addition, the authors are also encouraged to reference “SpecialtyScribe: Enhancing SOAP note Scribing for Medical Specialties using LLMs,” which demonstrates the application of LLMs in domain-specific knowledge editing and contextual reasoning. Including this citation would further emphasize the practical relevance of ThinkEval for evaluating editing robustness in specialized, high-stakes domains.

---

> ### Author Response · Authors · 2025-10-15
> **Official Comment by Authors**
>
> Dear Reviewer,
>
> We sincerely thank you for the time and effort spent reviewing our work and for providing such thoughtful and constructive feedback. Your comments have helped us improve the clarity and completeness of our work. We have carefully addressed each point in detail below.
> We respond to the **Weaknesses (W#)** and **Requested Changes (RC#)** as listed in your review:
>
> ---
> ### **W1 & RC1:**
>
> We appreciate the reviewer’s concern regarding computational efficiency and scalability. The majority of ThinkEval’s computational cost arises from CoT prompting. To better quantify this, we will measure the average end-to-end runtime of a complete ThinkEval process to generate 100 reasoning chains from a sample and report detailed statistics on GPU hours and human validation time, along with the scaling behaviour observed. Preliminary profiling indicates that CoT prompting contributes over 80% of the total runtime, while the validation and graph synthesis stages remain lightweight and can be effectively parallelized.
> While human involvement in the current process is already minimal, limited primarily to triplet verification, we will further automate earlier stages using an LLM-judge for initial triplet validation. To evaluate this trade-off, we will conduct an experiment on the 100-chain sample, comparing efficiency and accuracy with and without the LLM-judge. We will include quantitative results in the revised version.
>
> ---
> ### **W2 & RC2:**
> We agree that knowledge gaps across models can affect benchmark universality. As discussed in Appendix A, this represents a systemic limitation of model-agnostic evaluation rather than a flaw specific to KnowGIC. To assess the extent of this gap, we evaluated the original Qwen2.5-7B model on three datasets: **MQuAKE [1] (1000 sampled triplets): 85.5%**, **Know-MRI (known_1000 subset [2]): 84.4%**, and **KnowGIC: 81.0%**. These scores indicate that KnowGIC’s factual coverage is consistent with existing benchmarks. The purpose of Appendix A is precisely to highlight that perfect benchmark universality is inherently challenging, as factual recall varies across LLMs.
>
> KnowGIC was constructed using Llama3-8B primarily to demonstrate ThinkEval’s graph-based reasoning and indirect fact recovery analysis, rather than to claim model-agnostic completeness. Importantly, ThinkEval is model-adaptive and can rebuild or refine an existing dataset for any target model through a few iterative cycles. To illustrate this, we reconstructed a 100-path sample for *(Harry Potter, studied at, Hogwarts)* using **Qwen2.5-7B** within only five iterations of our ThinkEval process. Adapting from the existing KnowGIC sample would require even less time. We will include this sample in our anonymized repository alongside the original KnowGIC dataset.
>
> Regarding the reviewer’s suggestion for score normalization, both IFR and Preservation metrics inherently perform calibration by taking the pre-edit model performance (i.e., $R_{S_{i}}$ and $p_{t}$) as denominators in Equations (6) and (7), effectively normalizing for each model’s baseline factual accuracy before computing post-edit ratios. We realize that this normalization mechanism may not be immediately clear from the current text and will revise the manuscript accordingly.
>
> ---
> ### **W3:**
> Our current evaluation focuses on parameter-modifying techniques to enable consistent comparison under identical editing conditions (same model weights, prompt templates and evaluation metrics). This was intended to isolate the effects of internal parameter updates on indirect knowledge leakage, before extending to other approaches. We are running experiments on IKE and will provide the results very soon.
>
> ---
> ### **W5 & RC4:**
> We sincerely thank the reviewer for this very thoughtful question. Kudos to the reviewer. It highlights an exciting direction for future work. Our current work focuses on factual edits, as they offer a well-defined basis for evaluating indirect knowledge leakage. However, the underlying design of ThinkEval (its use of CoT reasoning and graph-based causal tracing) may be extensible to other forms of knowledge, including commonsense, procedural or rule-based reasoning. In such cases, nodes can represent abstract concepts or actions rather than entities, and the same CoT-driven triplet extraction process can capture causal or “if–then” relationships. We will include a brief discussion of this potential generalization direction in the revised version.
>
> ---
> ### References
> *[1] Zhong, Zexuan, et al. "Mquake: Assessing knowledge editing in language models via multi-hop questions." arXiv preprint arXiv:2305.14795 (2023).*
>
> *[2] Liu, Jiaxiang, et al. "Know-MRI: A Knowledge Mechanisms Revealer&Interpreter for Large Language Models." arXiv preprint arXiv:2506.08427 (2025).*

---

> ### Author Response · Authors · 2025-10-15
> **Official Comment by Authors**
>
> ### **W4:**
> We agree that the IFR metric could benefit from a clearer walkthrough. A working example is already provided in our anonymized repository, and we will include a detailed, illustrative example and visual schematic in our revised version.
>
> Below is a representative example illustrating how IFR is computed for a 3-step reasoning chain.
>
> **Edit to be made:**
> > *Harry Potter studied at Hogwarts → Ilvermorny.*
>
> **Sample reasoning chain:**
> 1. Who were Harry Potter's schoolmates? → Ron Weasley
> 2. Which house is Ron Weasley in? → Gryffindor
> 3. Which school does Gryffindor belong to? → Hogwarts
>
> **Chain Probabilities**
> | Question                                      | Pre-Edit | Post-Edit |
> | ----------------------------------------------- | -------- | --------- |
> | C1Q1: Who were Harry Potter's schoolmates?      | 0.90     | 0.70      |
> | C1Q2: Which house is Ron Weasley in?            | 0.85     | 0.80      |
> | C1Q3: Which school does Gryffindor belong to?   | 0.90     | 0.85      |
> **IFR Calculation**
>
> Pre-edit and Post-edit chain recovery:
> $$R_{pre} = 0.9 \times 0.85 \times 0.9 = 0.69, R_{post} = 0.7 \times 0.8 \times 0.85 = 0.48$$
>
> $$
> \text{IFR} = \frac{(R_{\text{post}} / R_{\text{pre}}) / \sqrt{n}}{(1 / \sqrt{n})}
>            = \frac{\frac{0.48}{\sqrt{3}}}{\frac{0.69}{\sqrt{3}}}
>            \approx 0.70
> $$
>
>
>
> This example demonstrates that even after editing, the original fact still remains 70% recoverable w.r.t the original model. Each link’s probability reflects how confidently the model reproduces the correct intermediate fact. The product of these probabilities gives the chain recovery, representing the overall likelihood that the original fact can be reconstructed through reasoning.
>
> The $( \frac{1}{\sqrt{n}} )$ weighting normalizes for chain length by mitigating the compounding uncertainty of multi-step reasoning.
> This prevents long, noisy chains from dominating the metric and emphasizes shorter, higher-confidence reasoning paths. We will refine the main-text explanation to make this intuition explicit.
>
> ---
> ### **W6, W7, W8, RC3 & RC5:**
> We agree that human validation may introduce potential bias and that reporting inter-annotator agreement is important. To address this, we have initiated a pilot annotation study with three of our research group members and will compute formal agreement statistics, reporting Gwet’s AC1 as the primary reliability metric alongside summary agreement statistics. We will also release anonymized disagreement logs to enhance transparency and trust in the evaluation process.
>
> In the revised version, we will include a subsection on responsible model editing, outlining ethical use cases, potential misuse scenarios and mitigation strategies such as transparent edit logging and human verification of sensitive changes. We have already included our prompt templates in Appendix E. We will release detailed implementation metadata, covering model versions and the hyperparameters used.
>
> We appreciate the suggestion to discuss mitigation strategies for achieving low IFR and high Preservation simultaneously. We will include a short discussion in the revised version outlining potential directions for future editing techniques that balance factual precision with contextual retention, positioning ThinkEval as not only an evaluator but also a guide for developing more robust and responsible editing methods.
>
> ---
> ### **RC6:**
> We appreciate this insightful suggestion that positioning ThinkEval more explicitly within the broader literature on editing robustness and consistency preservation will strengthen the theoretical grounding of our work. In the revised version, we will incorporate citations to the suggested studies. [3] and [4] provide valuable perspectives on mitigating catastrophic forgetting and maintaining representational consistency during model adaptation, which conceptually align with ThinkEval’s focus on preserving factual coherence and minimizing IFR degradation post-editing. Additionally, we will reference [5] to highlight the practical applicability of ThinkEval in domain-specific editing contexts. This inclusion will help emphasize how ThinkEval’s evaluation framework can inform robustness assessment in specialized, high-stakes applications such as clinical or technical domains.
>
> ---
> ### References
> *[3] Gao, Jianqi, et al. "Mitigating Forgetting in Adapting Pre-trained Language Models to Text Processing Tasks via Consistency Alignment." Proceedings of the ACM on Web Conference 2025. 2025.*
>
> *[4] Gao, Jianqi, et al. "Shaping pre-trained language models for task-specific embedding generation via consistency calibration." Neural Networks (2025): 107754.*
>
> *[5] Goyal, Sagar, et al. "SpecialtyScribe: Enhancing SOAP note Scribing for Medical Specialties using LLM’s." Proceedings of the Second Workshop on Patient-Oriented Language Processing (CL4Health). 2025.*

---

> ### Author Response · Authors · 2025-11-03
> **Summary of Experiments and Validation Results**
>
> *Note: All experiments are performed on an NVIDIA A100 80GB GPU.*
> ### **W1 & RC1**
>
> We measured the average end-to-end runtime of a complete ThinkEval process over incremental iterations of increasing sample sizes until at least 100 samples were generated.
>
> #### **Initial Setup:**
> Triplets: 1  Paths: 1
>
> | **Iteration** | **Triplets** | **Paths (Subject → Target)** | **CoT Queries per Iteration** |
> |----------------|---------------|------------------------------|-------------------------------|
> | 1 | 5  | 1   | 1  |
> | 2 | 15 | 3   | 4  |
> | 3 | 23 | 10  | 12 |
> | 4 | 51 | 27  | 40 |
> | 5 | 83 | 133 | 65 |
>
> **Average runtime per CoT prompt:** 4.15 seconds
> **Estimated total CoT GPU time:** ~8.4 minutes
> **Overall GPU runtime for 133 samples:** ~11 minutes
> **Human verification effort:** ~4–5 minutes
> **Total end-to-end time:** ~15 minutes
>
> These results confirm that while CoT prompting dominates computational cost, the validation and graph synthesis stages remain lightweight and can be effectively parallelized. We will include detailed results from this scalability study in the revised version.
>
> To further reduce human dependency, we implemented an LLM-based judge (using gpt-4o) that validates and prunes triplets at each stage, filtering misinformation and retaining only those useful for graph construction. The judge was evaluated on the same sample and assessed on its triplets classification for true positives (TP), false positives (FP) and false negatives (FN):
>
> | **Iteration** | **TP** | **FP** | **FN** | **Correct Paths Identified** |
> |----------------|---------|---------|---------|------------------------------|
> | 1 | 5  | 0  | 0  | 1   |
> | 2 | 13 | 1  | 2  | 3   |
> | 3 | 20 | 4  | 3  | 8   |
> | 4 | 44 | 8  | 7  | 23  |
> | 5 | 71 | 10 | 12 | 109 |
>
> The LLM-judge achieved an overall precision of 88%, recall of 86%, and F1-score of 87% in identifying valid triplets and reasoning paths (TP, FP and FN as 71, 10 and 12 respectively). Using the LLM-judge, ThinkEval reached the threshold of 100 reasoning chains within just five iterations, matching the case where only human verification was utilised. These findings suggest that with minor prompt-level refinements, the LLM-judge can reduce human validation effort by **~70–75%** with only marginal loss in accuracy, if human effort is reserved for final verification.
>
> ---
>
> ### **W3**
> We evaluated the parameter-preserving IKE technique [6] on the KnowGIC dataset, and the results are summarized below:
>
> | **Model** | **Preservation** | **IFR** |
> |------------|------------------|---------|
> | GPT2-XL | 0.680 | 0.542 |
> | Llama3-8B | 0.729 | 0.756 |
> | Qwen2.5-7B | 0.623 | 0.247 |
>
> These results indicate that indirect fact recovery can still occur even under parameter-preserving editing setups, suggesting that such techniques do not fully prevent hidden factual retention.
>
> ---
> ### **W6 & RC5**
> The implication chains in KnowGIC imply but may not directly assert a fact. Also, the strength of implication may decrease with chain length. Hence, we scale by **1/√n**, where *n* is the number of links, ensuring longer chains have proportionally reduced weight.
>
> 250 sample chains were reviewed by three members of our research group. Despite minor ambiguities, they consistently agreed on the presence of implication, and all agreed upon the √n scaling approach.
>
> All the factual triples were reviewed as well, and the results are as shown:
> | Metric            | Value |
> | :---------------- | :---- |
> | **Raw Agreement** | ~97%  |
> | **Gwet’s AC1**    | ~98%  |
>
> These values demonstrate strong consistency and minimal bias.
>
> ### **Noted Ambiguities (No Disagreement)**
>
> * Disney’s Nine Old Men: Some names correspond to multiple individuals (e.g., less-known persons sharing the same names), which could confuse the model. These ambiguous cases were already excluded from KnowGIC when it was uploaded, so the dataset isn't impacted by this at all.
> * Harry Potter samples: A minor canonical variation that Professor Quirrell taught Muggle Studies (in books only) and before Defence Against the Dark Arts. Such differences may occur in fictional setups.
>
> ### **Disagreements**
>
> 1. Familial Relationships: Disagreement on whether step-siblings should count as siblings in a few samples, as it may confuse the model.
> 2. YMCA Training School Location: Disagreement arose regarding the “original” institution in Springfield, Massachusetts versus its current global presence.
>
> ---
>
> ### References
>
> *[6] Zheng, Ce, et al. “Can We Edit Factual Knowledge by In-Context Learning?” Proceedings of the 2023 Conference on Empirical Methods in Natural Language Processing 2023.*

---

### Review · Reviewer_jLqF · 2025-10-31

**Summary Of Contributions:**

This paper focuses on Model Editing on LLMs - specifically, they attempt to study the phenomena of indirect knowledge leakage (also known as the "ripple effect") in case of such models. The basic idea is - once an LLM is edited with new information, can it reflect that new information in response to queries that have only indirect relation to such information? For that, they present ThinkEval - a novel framework to systematically evaluate such model editing techniques. This framework involves a detailed approach of building a knowledge graph using entity-relationship triplets, identifying sequences in such knowledge graphs, which are developed using Chain-of-thought reasoning from LLMs followed by "fact segmentation". They introduce Deep Editing to quantify indirect knowledge leakage through multi-step reasoning, and propose a metric called Indirect Fact Recovery (IFR) tailored to this setting. They use ThinkEval to develop the KnowGIC dataset of 1,406 multi-step sequential-prompting samples, which they use for evaluating combinations of LLMs and model editing techniques using the deep editing evaluation

**Additional Comments:**

NA

**Audience:**

Yes

**Audience Explanation:**

LLM evaluation is a super-active area of research. Model editing is one of the few new topics in this field, and ThinkEval provides a systematic framework to evaluate model editing techniques by quantifying the indirect propagation of knowledge. Additionally, they also contribute a benchmark dataset for this purpose. Hence, the LLM researchers in the TMLR audience will definitely be interested. In fact, researchers working on Knowledge Modeling will be interested to understand the entity-relation knowledge graph and associated algorithms presented here,

**Claims And Evidence:**

Yes

**Claims Explanation:**

The paper systematically describes the components of ThinkEval to systematically explain the knowledge graph structure, and present clear algorithms for query evaluation and refinement, automated triplet generation and developing chains, and quantify the amount of knowledge leakage by comparing the knowledge graph structure before and after model editing. They illustrate these concept with many illustrative examples from various domains. Experiments on LLM+editing technique comparison using IFR are very clearly explained.

**Requested Changes:**

I am generally satisfied with the paper, and have no changes to suggest. However, most of the examples illustrated in the paper are related to popular culture. Sometimes, query generation and knowledge propagation may be harder in scientific domains. I would love to see the how model editing works, and how ThinkEval can evaluate its performance on scientific fields like Physics, or even Machine Learning.

---

> ### Author Response · Authors · 2025-11-03
> **Official Comment by Authors**
>
> We sincerely thank the reviewer for the positive and encouraging feedback. We are glad that the reviewer found the contributions of ThinkEval clear, systematic and relevant to the LLM evaluation and model editing community.
>
> The reviewer's observation about extending ThinkEval to scientific domains such as Physics or Machine Learning aligns closely with the generalization discussion we addressed in **W5 & RC4** of Reviewer jMMQ. As noted there, while our current work focuses on factual edits, the underlying design of ThinkEval (its use of CoT reasoning and graph-based causal tracing) can extend to other forms of knowledge, including scientific, procedural or rule-based reasoning, with a few modifications. In such cases, nodes in the knowledge graph can represent abstract concepts, relations or operations rather than entities, allowing the CoT-driven triplet extraction process to capture causal and conceptual relationships in specialized domains. Datasets such as OpenBookQA[1] and S2ORC[2] can serve as valuable sources of base facts for this extension.
>
> We will include a brief discussion of this generalization in the revised version and plan to explore domain-specific reasoning in future extensions of ThinkEval.
>
> ---
>
> ### References
>
>
>
> *[1] Mihaylov, Todor, et al. “Can a Suit of Armor Conduct Electricity? A New Dataset for Open Book Question Answering.” *Proceedings of the 2018 Conference on Empirical Methods in Natural Language Processing*, Association for Computational Linguistics, 2018, pp. 2381–2391.*
>
> *[2] Lo, Kyle, et al. “S2ORC: The Semantic Scholar Open Research Corpus.” *Proceedings of the 58th Annual Meeting of the Association for Computational Linguistics*, Association for Computational Linguistics, 2020, pp. 4969–4983.*

---

### Review · Reviewer_B1vK · 2025-11-18

**Summary Of Contributions:**

**Summary:**
- The authors consider the complex task of selectively modifying facts after LLM training without costly re-training from scratch. The authors highlight that this task is difficult because editing out specific facts does not preclude the ability of a user to recover the removed fact using multi-step logical inferences, given access to the model; the authors have an example about recovering information about where Harry Potter goes to school, which they reference throughout the text and incorporate into their dataset.
- The authors propose formalizing the set of deductions as the set of all paths in a knowledge graph G, where nodes represent entities and edges represent relationships between entities. The intuition they propose is that “traversing [the edges of such a graph] naturally implies relationships between entities.” Thus, they propose that the set of all paths in G “captures all explicit and implicit relationships derivable from G through its paths.”
- The authors propose a framework for evaluating the effectiveness of different model editing techniques, which they call ThinkEval. ThinkEval is a meta-algorithm, in which chain-of-thought prompts are used to generate knowledge graphs modeling a set of relationships known to an LLM. The LLM is then modified using the editing technique under investigation. After editing, the knowledge graphs generated pre-edit can then be used to generate multi-hop queries, to assess whether the edited model can actually be prompted to recall the selected deleted fact(s) based on other relationships retained within the knowledge graph.
- The authors formulate “deep editing” as a new target task. Unlike previous editing task definitions, deep editing is only considered successful if an entity-relationship-entity triplet does not belong to the closure of the knowledge graph of the LLM after editing.
- In order to quantify deep editing performance, the authors introduce Indirect Fact Recovery (IFR). This metric computes the weighted average of retained path lengths, normalized to place a higher penalty on shorter paths.

**Strengths:**
The authors tackle a very ambitious and relevant problem. Even their flagship example (Harry Potter) calls attention to the extreme importance of model editing in pursuit of respecting copyright.
The specific concern about recovering edited facts using multi-step logical inference is also very well-motivated, as the authors point out how brittle targeted fact-forgetting can be.

**Weaknesses:**
- The proposed framework (ThinkEval) seems difficult to automate and thus to scale.
   - Consider the first step, “Query Validation and Refinement.” This requires turning a triplet or chain of relationships into a natural language query. The natural language query requires specific background knowledge to formulate. In their example, the authors state that (Harry Potter, subject, Transfiguration, taught by, Minerva McGonagall) becomes “Who teaches Transfiguration to Harry Potter?” It seems that generically triplets/chains may have many interpretations: why not “By whom was Harry Potter subjected to Transfiguration, as taught by Minerva McGonagall?” a prompt with the same entities/relationships but which is nonsensical and does not align with background knowledge from the Harry Potter series. This causes me to question how effectively the process of converting triplet/chains into prompts can be automated.
   - How do the authors propose to automate such a process? My understanding from the manuscript is that in practiece they deploy a secondary LLM agent for converting triples/chains into prompts, and which then “analyzes [the target model’s response] to determine the expected object’s presence.” Is this correct?
   - At other steps, the authors explicitly acknowledge that human users will be required to perform large amounts of work as part of ThinkEval. At a minimum, the proposed framework requires that human users manually (a) refine queries/triplets that fail, (b) verify and prune triplets propuced during Automated Triplet Generation.
- Changing any part of the ThinkEval meta-algorithm (e.g., the relative workload of human versus LLM agents in generating and refining queries) seems like it would significantly change important aspects of the evaluation. Thus it seems difficult to interpret any comparison between results obtained by different groups when using ThinkEval, which seems to limit its applicability. Do the authors see any avenues that would mitigate this difficulty?
   - E.g., even given the same set of base triplets, the procedure of query generation/refinement involves significant qualitative steps, deciding when responses constitute “authentic triplets” versus misinterpretations.
- Overlap with existing benchmarks: the proposed benchmark framework has significant overlap with MQuAKE, but the authors include minimal comparison with MQuAke and/or motivation for the creation of a new knowledge leakage framework.
   - Do the authors have any examples of model editing techniques for which ThinkEval detects knowledge leakage but MQuAKE does not? Does ThinkEval suggest any new facts about the set of editing techniques evaluated compared to evaluation with MQuAKE? At least two of the techniques evaluated (MEMIT and ROME) also seem to be evaluated in the MQuAKE initial manuscript, so the authors could comment on comparisons of those methods without needing to run new experiments.
   - The authors claim that “Though [MQuAKE is] valuable for probing broader knowledge, it doesn’t consider scenarios where unmodified connected facts logically reconstruct the original fact.”
   - However, the example on the first page of MQuAKE (British Prime Minister) seems analogous to the Harry Potter example. I.e. the model’s failure in incorrectly responding “Carrie Johnson” when asked “Who is married to the British Prime Minister” seems analogous to answering “Draco Malfoy” as part of the Harry Potter knowledge leakage reasoning chain. I.e. one could consider the fact that MQuAKE reveals that “The Brittish Prime Minister is married to Carrie Johnson” as a scenario where the model [erroneously] reconstructs that “Borris Johnson is the British Prime Minister.” Thus MQuAKE seems very conceptually related to ThinkEval, and it is not clear what benefit the authors propose over MQuAKE.

**Questions:**
- See my questions about comparisons with MQuAKE in “Weaknesses.”
- See my questions about the difficulty of comparing ThinkEval results across different research groups.
- How stable are ThinkEval results to the choice of base-query datasets? The knowledge graph of modern LLMs is so massive that I’m concerned about whether it is feasible using this human-in-the-loop procedure to sample a sufficiently large fraction of the knowledge graph as to be able to reliably compare different editing techniques.
- I’m confused about the validity of the intuition: “G is designed such that traversing its edges naturally implies relationships between entities.” This doesn’t seem generically true unless “relationships” are constrained to have transitive properties. E.g. the path (Snake, predator, mouse, smaller than, elephant) doesn’t imply any relationship between snake and elephant. Is the idea that the closure of G is a superset of all reasonable relationships, even if it includes vacant/erroneous relationships?

**Additional Comments:**

I find it misleading to refer to ThinkEval as an “architecture” in Fig. 2. ThinkEval is a framework or meta-algorithm, not a parameterized model.

Typos:
In the opening sentence of the abstract, writing “to enable…” feels grammatically awkward. I might suggest “as they enable…”
Page 5, grammatical error: “For users, choosing the right model-editing technique for their model is vital by evaluating…”

**Audience:**

Yes

**Audience Explanation:**

The task identified by the authors of ensuring that edits to LLMs are robust to chains of logical inference is extremely challenging and relevant.

**Broader Impact Concerns:**

I am unconvinced of the stability and replicability of the proposed evaluation framework. Thus, if this framework were deployed, I am concerned that it could yield misleading or unreplicable results to practitioners. Given the heightened current interest in LLMs and editing methods, this seems like a substantial potential for harm.

**Claims And Evidence:**

Yes

**Claims Explanation:**

The authors claim that they introduce a new framework (ThinkEval), a new editing task (deep editing), and a new evaluation metric (IFR). These claims are correct. I have concerns about the utility of the evaluation procedure that they introduce, but these are not correctness issues per se.

**Requested Changes:**

- The authors must include some comparison with other editing evaluation frameworks in their experiments. MQuAKE seems natural as it is conceptually related and because KnowGIC is built on MQuAKE’s dataset.
- The authors must make explicit which parts of their meta-algorithm they intend to be automated with LLMs. Currently, such as “turning base triplets into queries” are left ambiguous as to whether they are intended for human-in-the-loop work versus automation with yet another LLM.
- The authors need to discuss replicability and stability. To scale ThinkEval, it seems necessary to use a second LLM in many parts of the meta algorithm (query generation, query refinement, query validation, triplet extraction, triplet pruning, etc). It seems very difficult to control for the impact that this secondary LLM will have on the evaluation results. Empirically, how significantly do the results provided by ThinkEval change when different automation workflows are used? Is the relative ranking of editing methods (Figure 5) or counterintuitive phenomena (“rise-then-fall” in Figure 4) preserved under different automation workflows? How do the authors propose that the community compare different results obtained using ThinkEval as these workflows change?

---

> ### Author Response · Authors · 2025-11-21
> **Official Comment by Authors**
>
> Dear Reviewer,
>
> Thank you for the time and effort you devoted to evaluate our work, and for recognising the importance of recovering edited-out facts via multi-step logical inference in model editing. We have carefully considered each of your comments and provided detailed responses below. **Weaknesses (W#)**, **Questions (Q#)**, and **Requested Changes (RC#)**:
>
> ---
> ### **RC2**
>
> Thank you for the suggestion. `We clarify our entire process below and will refine it in the revised manuscript`. **No secondary LLM agent is used in ThinkEval.** We use only one unedited model.
> Automation Type|Component|Where in Figure 2|Comment
> -|-|-|-
> Using unedited original LLM|Query generation|Right before blue region ①|Similar to LLM-driven query generation in [1][5]
> ||LLM response generation|Blue region ①
> ||Chain-of-Thought answering|Pink region ②
> ||Triplets extraction|Pink region ②|Similar to [4] and other benchmarks
> No LLM or human involvement |Knowledge validation|Blue region ①|Rule-based, same way as [1] (using alias list)
> ||Fact segmentation|Pink region ②|Segmented into individual statements
> ||Chain sequencing|Green region ③
> Human-in-the-loop (HITL)| Query refinement|Blue region ①
> ||Triplet pruning and addition|Pink region ②
> ||Final sanity checks|Dataset (Green region ③)
>
> ---
> ### **W1 & Q3**
>
> Below, we show that ThinkEval is scalable and automatable, and human involvement is minimal. Our approach is consistent with existing evaluation frameworks and supported by empirical measurement. We provide the details for automation in the above response for **RC2**. As already mentioned, **ThinkEval doesn't use any secondary LLM agent.**
>
> **’Query Validation and Refinement’ step reliably automates triple/chain conversion into queries based on the following three points.**
>
> 1. Consistency with literature: Several benchmarks like MQuAKE generate queries directly from triples using LLMs. [1][2][3][4] use gpt-3.5-turbo and GPT-4o for multi-hop query generation, event-based edits generation, extensive paraphrasing and triples extraction. [5] uses various models to generate numerous rephrases of the same sample. [6] shows that LLMs can autonomously extract triples with minimal supervision.
> 2. Controlled query generation: ThinkEval’s prompt design (Appendix E) generates queries so that the expected answer is the final object of the triplet/chain, and the query itself never contains that object. The Knowledge Validation step further enforces this: a query is accepted only if the object is (1) excluded in the query, and (2) included in LLM’s response. These constraints prevent wrong formulations as the Reviewer rightly points out.
> 3. Annotator agreement: As shown in our response to Reviewer jMMQ’s W6 & RC5, all triples and their generated queries are reviewed by three non-author annotators in our human-agreement study. We obtain a raw agreement of ~97% and Gwet’s AC1 of ~98%, indicating that the generated queries are consistently interpretable. For transparency, we release the disagreement logs in the aforementioned response.
>
> **ThinkEval is stable in terms of human involvement.** As discussed in Appendix B, limited human-in-the-loop (HITL) is standard practice in LLM evaluation and model editing [2][3][8]. Due to hallucinations, lightweight human verification remains crucial. Because knowledge is ever-evolving, it is extremely difficult to empirically establish any knowledge graph as complete [9][10]. The core objective of ThinkEval is not to obtain complete knowledge graphs but to operationalize deep editing (Section 5).
>
> In our response to Reviewer jMMQ’s W1 and RC1, we ran a scalability study using our full pipeline to generate 100 chains from one base triple. Results:
>
> - GPU runtime: ~11 min
> - Human time: ~4–5 min
> - End-to-end runtime: ~15 min
>
> Thus, producing 100 samples required only ~4–5 minutes of human effort.
>
> **Optional enhancement for improved scalability.** A secondary LLM is not part of our core pipeline. As recommended by Reviewer jMMQ (RC1), we experimented with a stronger "judge" LLM (GPT-4o) for initial triple validation and pruning. This reduces human effort by 70–75%, leaving only final verification steps to humans, so that dataset quality doesn’t drop. Existing literature [7] also uses LLMs to validate and correct extracted knowledge-graphs with minimal human interference.
>
> Thus, human involvement in ThinkEval is already minimal, and an LLM-judge further decreases it. `We will incorporate these results into our revised manuscript`. ThinkEval’s evaluation is model-adaptive, as all reasoning chains originate from the unedited model. Even when a secondary LLM is used, evaluation stays grounded for the target model.

---

> > ### Author Response · Authors · 2025-11-21
> > **Official Comment by Authors**
> >
> > ---
> > ### **W2, Q2, RC3 & Broader Impact Concern**
> >
> > We thank the Reviewer for their comments. While we acknowledge the Reviewer’s concerns about reproducibility and stability issues with LLM usage, **it has become a standard and important practice to use LLMs in creating editing benchmarks.** [1][2][3][4][5] (including MQuAKE) rely heavily on LLMs to prepare datasets, and for editing and unlearning. [12] evaluates LLMs as judges and reports high agreement with human ratings. [1][2][11][13] utilise paraphrasing to evaluate generalization. This introduces controlled surface-level variation, which is **rather beneficial, not harmful**, as long as there is controlled generation of samples. We ensure this consistency via controlled query generation and HITL verification (as noted above in W1).
> >
> > **While we have not used secondary LLMs in ThinkEval, using an LLM-as-a-judge in ThinkEval is recommended by Reviewer jMMQ (W1 & RC1)** for filtering triples (reserving human effort for final verification). As shown in response to the aforementioned review, the LLM-judge achieved an F1-score of 87% in identifying valid triples and paths. A final human verification minimises human effort, ensuring that all samples are valid.
> >
> > `To address Reviewer’s question about the stability of technique rankings, we will paraphrase all queries in the 100-path graph from our scalability study (response to Reviewer jMMQ’s W1 and RC1) and evaluate all editing techniques. We will report shortly in a few days whether the ranking of editing techniques and the rise-then-fall pattern remain preserved.`
> >
> > **Public release of resources:** To ensure cross-group consistency, we already include deterministic constraints in query generation, validation and HITL verification (as noted in W1). We have already released: ① our dataset, and ② all prompt templates (Appendix E). `We will additionally release all model versions and hyperparameters.` Practitioners can directly use these resources, and we recommend them to stick to the given pipeline and release datasets for reproducibility.
> >
> > ---
> > ### **W3 & Q1**
> > We thank the Reviewer for highlighting this point. We list the differences below to show that ThinkEval and MQuAKE are complementary, not overlapping.
> > Dimension|MQuAKE|ThinkEval
> > -|-|-
> > Fundamental difference|Dataset|Framework to create datasets
> > What they evaluate|Multi-hop reasoning in edited LLMs; however, multi-hop reasoning remains beyond current parameter-editing techniques, which still struggle with ripple effects [14][16][17]|Shows edited-out facts leakage via sequential queries; bridges gap between reasoning and ripple effects by mapping paths in knowledge graphs; thereby **highlighting which additional facts require editing and which must be preserved**
> > How they evaluate|Multi-hop reasoning questions where multiple relations are compressed into a single query|Multi-step sequential queries; checks reasoning across sequential queries, extending evaluation beyond single-query formats, to reflect how users may naturally query LLMs
> > Adaptability|① As shown in response to Reviewer jMMQ’s W2 & RC2, unedited Qwen2.5-7B correctly answers ~85.5% of base triples; ② Unedited models correctly answer only 30–40% of MQuAKE’s multi-hop queries (as per their own results [1]), restricting its usable portion|Resolves ① by adapting dynamically to any model; Multi-step inference mitigates ② by decomposing reasoning into simpler sequential steps
> > Metrics|Single-query metrics|IFR (Section 5.1); Preservation [14]
> >
> > **Our results reveal a consistent trade-off in editing techniques between indirect fact suppression and preservation** (Table 4, Figure 6), which single-query format samples cannot fully reveal. Moreover, unedited models answer only 30–40% of MQuAKE’s multi-hop questions correctly (as in point 4 above [Adaptability]). ThinkEval mitigates this by decomposing reasoning into atomic, sequential steps. As MQuAKE evaluates a different set of LLMs, `we will run an experiment to compare results under both settings within a few days.`
> >
> > **Different examples in ThinkEval and MQuAKE.** The MQuAKE example ("Who is married to the British Prime Minister?") illustrates a 2-hop generalization failure due to outdated knowledge not being updated. ThinkEval examines a different phenomenon: whether an LLM can leak an edited-out fact through a sequence of queries ("Who is Harry Potter's schoolmate?" → "Where did Draco Malfoy study?"). As noted in point 3 [How they evaluate], MQuAKE does not test sequential recovery of facts, which is how users may naturally query LLMs. As noted in point 2 [What they evaluate], we explicitly reveal, via knowledge graphs, which related facts should also change and which must be preserved. ThinkEval does not replace MQuAKE. It fills a critical gap by measuring hidden inference pathways through which edited-out facts can still leak.
> >
> > We thank the Reviewer for highlighting this ambiguity in our phrasing. `We will refine this in the revised manuscript.`

---

> > > ### Author Response · Authors · 2025-11-21
> > > **Official Comment by Authors**
> > >
> > > ---
> > > ### **RC1**
> > > A comparison of our editing setting with other editing settings [2][3] is already provided in Appendix C. Appendix H contrasts our IFR metric with prior editing metrics [13][14]. While KnowGIC uses MQuAKE’s base triples as a seed set, this mirrors how MQuAKE itself uses base triples from [15], and how [2] does from [13]. ThinkEval does not rely on MQuAKE’s multi-hop questions, templates or reasoning data. `We will incorporate the comparison between ThinkEval and MQuAKE discussed in W3 into our updated manuscript.`
> > >
> > > ---
> > > ### **Q4**
> > > **Each path in our graphs is meant to imply a meaningful relationship between the base subject and object.** All chains undergo a final sanity check by the authors. Any chain plausibly not implying the subject-object relationship (like the Reviewer’s example) is discarded for indirect fact recovery. Its individual queries may still be retained for ripple-effect evaluation. As noted in our response to Reviewer jMMQ’s W6 & RC5, three non-author annotators in our human-agreement study reviewed 250 chains. With minor ambiguities and zero disagreements, they consistently approved their correctness. We also provide the logs in that response.
> > >
> > > This HITL verification step aligns with established practice in model editing datasets [2][3][4]. Thus, we don’t treat the closure of G as a blanket set of implied facts. Rather, G is filtered such that its chains plausibly and coherently imply the target fact. `We will revise this in the updated manuscript for clarity.`
> > >
> > > ---
> > > ### **Regarding Additional Comments**
> > > We thank the Reviewer for these constructive suggestions. `We will revise Fig. 2 caption to refer to ThinkEval as a framework, and will correct the additional issues noted.`
> > >
> > > ---
> > > ### **References**
> > >
> > > *[1] Zhong, Zexuan, et al. "Mquake: Assessing Knowledge Editing in Language Models via Multi-hop Questions." Proceedings of the 2023 Conference on Empirical Methods in Natural Language Processing.*
> > >
> > > *[2] Liu, Jiateng, et al. "EVEDIT: Event-Based Knowledge Editing for Deterministic Knowledge Propagation." Proceedings of the 2024 Conference on Empirical Methods in Natural Language Processing.*
> > >
> > > *[3] Deng, Jingcheng, et al. "Everything Is Editable: Extend Knowledge Editing to Unstructured Data in Large Language Models." Proceedings of the 2025 International Conference on Learning Representations.*
> > >
> > > *[4] Wang, Changyue, et al. "Knowledge Editing through Chain-of-Thought." Proceedings of the 2025 Conference on Empirical Methods in Natural Language Processing.*
> > >
> > > *[5] Sinha, Yash, et al. "UnSTAR: Unlearning with Self-Taught Anti-Sample Reasoning for LLMs." Transactions on Machine Learning Research, 2025.*
> > >
> > > *[6] Ananya, Ananya, et al. "Towards Harnessing Large Language Models as Autonomous Agents for Semantic Triple Extraction from Unstructured Text." Extended Semantic Web Conference. 2024.*
> > >
> > > *[7] Boylan, Jack, et al. "KGValidator: A Framework for Automatic Validation of Knowledge Graph Construction." Proceedings of the 3rd International Workshop on Knowledge Graph Generation from Text (TEXT2KG) & Data Quality meets Machine Learning and Knowledge Graphs (DQMLKG), Joint Workshop co-located with ESWC, 2024.*
> > >
> > > *[8] Tsaneva, Stefani, et al. "Knowledge graph validation by integrating LLMs and human-in-the-loop." Information Processing & Management, 2025.*
> > >
> > > *[9] Yang, Haotong, et al. "Rethinking knowledge graph evaluation under the open-world assumption." Advances in Neural Information Processing Systems, 2022.*
> > >
> > > *[10] Peng, Ciyuan, et al. "Knowledge graphs: Opportunities and challenges." Artificial intelligence review, 2023.*
> > >
> > > *[11] Levy, Omer, et al. "Zero-Shot Relation Extraction via Reading Comprehension." Proceedings of the 21st Conference on Computational Natural Language Learning (CoNLL 2017).*
> > >
> > > *[12] Zheng, Lianmin, et al. "Judging llm-as-a-judge with mt-bench and chatbot arena." Advances in neural information processing systems, 2023.*
> > >
> > > *[13] Meng, Kevin, et al. "Locating and editing factual associations in gpt." Advances in neural information processing systems, 2022.*
> > >
> > > *[14] Cohen, Roi, et al. "Evaluating the ripple effects of knowledge editing in language models." Transactions of the Association for Computational Linguistics, 2024.*
> > >
> > > *[15] Vrandečić, Denny, and Markus Krötzsch. "Wikidata: a free collaborative knowledgebase." Communications of the ACM, 2014.*
> > >
> > > *[16] Qin, Jiaxin, et al. "Why does new knowledge create messy ripple effects in llms?." Proceedings of the 2024 Conference on Empirical Methods in Natural Language Processing.*
> > >
> > > *[17] Wang, J., et al. "The Missing Piece in Model Editing: A Deep Dive into the Hidden Damage Brought by Model Editing." ICASSP 2025: IEEE International Conference on Acoustics, Speech and Signal Processing, 2025.*

---

> > > > ### Author Response · Authors · 2025-11-25
> > > > **Summary of Additional Experiments**
> > > >
> > > > As stated in our earlier response, we conducted two additional experiments:
> > > > ### **1. Direct comparison between MQuAKE and our KnowGIC dataset (for W3 & Q1)**
> > > > We conducted a controlled experiment comparing ThinkEval (KnowGIC dataset) and MQuAKE using the same set of base triples and the same reference model (GPT-J-6B). The authors in MQuAKE originally filtered out multi-hop queries from their dataset which contained any atomic fact that GPT-J couldn’t recall (Appendix A.2 of MQuAKE). Thus, for the set of base triples shared between MQuAKE and KnowGIC, GPT-J recalls 100% of the underlying atomic base facts. Hence, we utilise GPT-J as the reference model for our comparison.
> > > >
> > > > We measured the proportion of samples that remain usable (i.e., correctly answered by original, unedited GPT-J) under each benchmark:
> > > > Framework|Usable portion of dataset for GPT-J
> > > > -|-
> > > > MQuAKE (multi-hop)|**54%**
> > > > (Ours) KnowGIC (multi-step)|**87%**
> > > >
> > > > Although both benchmarks start from the same set of base triples, the proportion of usable evaluation samples is substantially higher under ThinkEval. In MQuAKE, only 54% of multi-hop samples remain usable for evaluating on GPT-J, despite all component facts being individually recallable. This is consistent with the MQuAKE authors' own observations that their chosen unedited models answer only 30–40% of MQuAKE's multi-hop questions correctly (MQuAKE, Table 3).
> > > >
> > > > In contrast, ThinkEval's sequential decomposition produces simpler single-relational queries at each step, resulting in 87% usability. Thus, ThinkEval complements MQuAKE by adapting to a model's reasoning capabilities and enabling measurement of deep-editing leakage in settings where MQuAKE's multi-hop questions are inefficient in meaningfully probing model behavior.
> > > >
> > > > ---
> > > > ### **2. Automated-Workflow and Results Stability Test (for W2, RC3 & Broader Impact Concern)**
> > > > We conducted a controlled stability experiment to test whether ThinkEval’s findings **(editing-technique rankings, and the “rise-then-fall” behaviour in IFR) remain stable even when a secondary LLM is introduced into the pipeline.**
> > > >
> > > > **Experimental setup.** We use a 100-chain subset and employed GPT-4o (the secondary model) to generate three paraphrases for every atomic triple (e.g., (Harry Potter, school, Hogwarts) → ‘Harry Potter studied at’,’Harry Potter's school is’,’Harry Potter went to school at’). All five editing techniques were evaluated across three models (GPT-2-XL, Llama-3-8B, Qwen2.5-7B). To incorporate paraphrasing, each link in a chain was scored by averaging across its three paraphrased queries, and IFR and Preservation were recomputed accordingly.
> > > >
> > > > **We have uploaded the entire subset** with the three paraphrasings for each atomic triple **on our anonymised repository along with the resultant plots** ((I) IFR trends for different chain lengths, and (II) IFR vs Preservation) **for transparency and easier visualisation.**
> > > >
> > > > **Results:**
> > > > Model|Technique|IFR (↓)|Pres. (↑)|Overall|1-step|2-step|3-step|4-step|5-step
> > > > -|-|-|-|-|-|-|-|-|-
> > > > GPT2-XL|AlphaEdit|0.736|0.903|0.736|0.433|0.773|0.824|0.652|0.442
> > > > ||MEMIT|0.727|0.880|0.727|0.433|0.903|0.714|0.671|0.419
> > > > ||PRUNE|0.250|0.782|0.250|0.311|0.280|0.248|0.232|0.201
> > > > ||RECT|0.754|0.850|0.754|0.667|0.752|0.829|0.686|0.581
> > > > ||ROME|0.279|0.752|0.279|0.233|0.250|0.282|0.301|0.272
> > > > |Llama-3-8B|AlphaEdit|0.678|0.900|0.678|0.342|0.604|0.683|0.781|0.478
> > > > ||MEMIT|0.800|0.911|0.800|0.615|0.867|0.814|0.767|0.629
> > > > ||PRUNE|0.309|0.647|0.309|0.111|0.434|0.363|0.170|0.183
> > > > ||RECT|0.894|0.880|0.894|0.615|0.897|0.925|0.875|0.825
> > > > ||ROME|0.217|0.571|0.217|0.081|0.350|0.167|0.187|0.219
> > > > |Qwen2.5-7B|AlphaEdit|0.391|0.917|0.391|0.285|0.315|0.512|0.313|0.269
> > > > ||MEMIT|0.394|0.865|0.394|0.297|0.392|0.458|0.323|0.339
> > > > ||PRUNE|0.411|0.587|0.411|0.316|0.320|0.442|0.445|0.419
> > > > ||RECT|0.525|0.823|0.525|0.387|0.492|0.558|0.520|0.486
> > > > ||ROME|0.261|0.537|0.261|0.198|0.237|0.312|0.221|0.224
> > > >
> > > > **Observations.** Across all models and editing techniques, we observe high stability:
> > > > 1. **Relative ranking is preserved.** Across all three models, the relative ranking of editing techniques is quite similar to that observed in the original (non-paraphrased) evaluation. AlphaEdit, MEMIT, RECT consistently exhibit higher IFR but strong Preservation. ROME and PRUNE consistently yield low IFR but lower Preservation. This mirrors the same trade-off we reported originally.
> > > > 2. **The same characteristic rise-then-fall in IFR persists amidst paraphrasing.** This shows that the behaviour is not due to specific phrasing, but a stable pattern of how the edited models propagate information across multi-step inference.
> > > >
> > > > The stability of IFR, Preservation, and editing-technique rankings show that ThinkEval’s core findings are robust to changes in the meta-algorithm. Reasonable automation choices (like using a secondary LLM for paraphrasing) do not alter the qualitative conclusions. Practitioners can expect consistent outcomes across groups, even when workflows differ across groups.

---

> > ### Comment · Reviewer_B1vK · 2025-11-24
> >
> > I thank the authors for their thorough responses to my questions.
> >
> > I have one quick follow-up question about the triplet->query process. Given the authors' response, my understanding is now that the target model used in the editing procedure is used to develop queries based on triplets, i.e. given a triplet (entity_1, relationship, entity_2) the target model forms a query of the type "What entity [has relationship] to entity_1?" such that the target answer is entity_2. My only hesitation about such a procedure is that since the target model forms this query given entity_2 in the prompt triplet, it seems possible that the model will construct a query that has higher internal association with entity_2 than other natural language prompts generated by sources that do NOT have knowledge of the target model's internal association probabilities. E.g. in the Harry Potter example, maybe this model has much higher association of the word "schoolmate" with Draco Malfoy than "classmate." In the extreme case, one can imagine a scenario where some seemingly random word in the query prompt, not related to entity_1 or to [relationship], has a very strong internal probability of association with entity_2, and thus the model is able to reconstruct the answer not because of some preserved multi-hop reasoning but because this "codeword" appeared in the prompt. This seems like it could impact the probability of recovery after editing, potentially making it easier for the target model to reconstruct the edited-out fact than "typical" queries generated by external agents or by human users.
> >
> > Is this of any concern to the authors, or is this an innocuous dependency?

---

> > > ### Author Response · Authors · 2025-11-25
> > > **Response to Follow-up Question**
> > >
> > > Dear Reviewer, thank you for raising this question. We agree that, in principle, allowing the same model to generate queries and to answer those queries could introduce a “codeword leakage”. For example, the model might disproportionately associate certain lexical choices (“schoolmate” vs. “classmate”) with the target object (entity_2). We address this concern below.
> > >
> > > First of all, as noted in our response to **W1 & Q3**, we use strict constraints and a validation step to ensure that entity_2 doesn't appear in the generated query. **This mitigates the issue partially**. For example, for the triple (Harry Potter, school, Hogwarts), only queries such as  “Where did Harry Potter study?” are accepted, whereas queries like “Did Harry Potter study at Hogwarts?” are rejected (as entity_2 Hogwarts appears in the query).
> > >
> > > Furthermore, as shown in our previous response, we conducted an automated-workflow and results stability test, where we paraphrase all the atomic triples into three samples each by a secondary LLM (GPT-4o). **This breaks any reliance on the target model's internal lexical associations**. Across all three LLMs, we observe that relative rankings of the editing techniques remain similar, the IFR rise-then-fall pattern persists and IFR-Preservation trade-offs remain consistent. This indicates that our evaluation is not impacted by model-specific lexical choices.
> > >
> > > Although we do not observe the codeword effect in practice, **increasing the degree of paraphrasing** is indeed an effective mitigation strategy.

---

### Decision · Action_Editor_M4j5 · 2026-01-10

**Recommendation:** Accept as is

**Additional Comments:**

Based on the reviews and subsequent discussions, the decision is to accept the paper as is but please include any asked / promised changes in the camera-ready version.

**Audience:**

Yes

**Audience Explanation:**

The paper concerns with knowledge leakage in LLM for deep editing scenarios.

**Claims And Evidence:**

Yes

**Claims Explanation:**

The paper studies LLM editing and the associated issues. To this end, it presents the framework ThinkEval to systematically examine the effect of editing vis-a-vis multi-step or indirect information leakage. It evaluates a number of editing techniques and particularly focuses on LLM editing in the context of reasoning so that indirect leakage can be minimized. A core contribution is the KnowGIC dataset. It also proposes metrics specifically designed to address indirect information leakage. The claims are adequately supported by experiments and benchmarking.

The paper is very timely to the community.

The reviewers have liked the work. There were initial concerns about scalability, reproducibility which the authors have addressed in the revised version. Reviewers have appreciated the efforts, and overall, they are in favor of accepting the paper.